



# New insights into Rossby wave packet properties in the extratropical UTLS using GNSS radio occultations

Robin Pilch Kedzierski[1], Katja Matthes[1,2], and Karl Bumke[1]

[1]Marine Meteorology Department, GEOMAR Helmholtz Centre for Ocean Research Kiel, Kiel, Germany.
[2]Faculty of Mathematics and Natural Sciences, Christian-Albrechts-Universität zu Kiel, Kiel, Germany.

**Correspondence:** Robin Pilch Kedzierski (rpilch@geomar.de)

**Abstract.**

The present study describes Rossby wave packet (RWP) properties in the upper-troposphere and lower-stratosphere (UTLS) with the use of Global Navigation Satellite System radio occultation (GNSS-RO) measurements. This global study covering both hemisphere's extratropics is the first to tackle medium and synoptic-scale waves with GNSS-RO. We use one decade of

GNSS-RO temperature and pressure data from the CHAMP, COSMIC, GRACE, Metop-A, Metop-B, SAC-C and TerraSAR-X missions; combining them into one gridded dataset for the years 2007-2016. Our approach to extract RWP anomalies and their envelope uses Fourier and Hilbert transforms over longitude without pre- or post-processing the data. Our study is purely based on observations, only using ERA-Interim winds to provide information about the background wind regimes.

The RWP structures that we obtain in the UTLS agree well with theory and earlier studies, in terms of coherent phase/group

propagation, zonal scale and distribution over latitudes. Furthermore, we show that RWP pressure anomalies maximize around the tropopause, while RWP temperature anomalies maximize right above tropopause height with a contrasting minimum right below. RWP activity follows the zonal-mean tropopause during all seasons.

RWP anomalies in the lower stratosphere are dynamically coupled to the upper troposphere. They are part of the same system with a quasi-barotropic structure across the UTLS. RWP activity often reaches up to 20 km height and occasionally

higher, defying the Charney-Drazin criterion. We note enhanced amplitude and upward propagation of RWP activity during sudden stratospheric warmings.

We provide observational support for improvements in RWP diagnostics and wave trend analysis in models and reanalyses. Wave quantities follow the tropopause, and diagnosing them on fixed pressure levels (which the tropopause does not follow) can lead to aliasing. Our novel approach analysing GNSS-RO pressure anomalies provides wave signals with better continuity

and coherence across the UTLS and the stratosphere, compared to temperature anomalies. Thus, RWP vertical propagation is much easier to analyse with pressure data.



# 1 Introduction

Rossby wave packets (RWPs) are transient and zonally confined undulations of the extratropical westerly flow. They can be seen as an organized succession of troughs and ridges with subplanetary zonal scale and a characteristic time scale from days to a couple of weeks (Wirth et al., 2018). The dispersive nature and downstream development of RWPs has long been noted (Rossby, 1945; Hovmöller, 1949) due to their faster group speed than phase speed (Andrews et al., 1987). RWPs propagate
along 'waveguides', narrow bands of sharp isentropic potential vorticity (PV) gradients (Hoskins et al., 1985; Shapiro and Keyser, 1990; Martius et al., 2010) and are fuelled mainly by baroclinic energy generation (e.g., Chang, 2001), thereby also being referred to as baroclinic waves. RWPs are steered by the polar front jet, reinforcing it via wave-mean flow interaction (Hoskins et al., 1983), and they represent transient states of the storm tracks, regions with frequent cyclone occurrence (Hoskins
and Valdes, 1990; Chang et al., 2002).

The horizontal scale of RWPs varies seamlessly between medium (wavenumbers w4-7) and synoptic scale (w > 7), therefore an intermediate range of w4-15 or similar is generally used to study RWPs (e.g., Zimin et al., 2003; Glatt and Wirth, 2014; Wolf and Wirth, 2017; Quinting and Vitart, 2019). The extent in longitude of RWPs, or the size of the envelope that modulates the wave's amplitude, also shows high variability: in the Southern hemisphere (SH) a more global scale is observed while in
the Northern hemisphere (NH) the waves tend to be more localized, due to the extent and zonality of the waveguide (e.g., Lau, 1979; Randel and Stanford, 1985). Nevertheless, either behaviour can occur in both hemispheres at specific times and even when RWPs reach a global scale, the amplitude of the wave is rarely constant along all longitudes. Global or local wave modes show no apparent differences in their dynamics (Randel and Stanford, 1985). For consistency we will use the term RWP throughout the manuscript to refer to wave activity, since it simultaneously covers the medium and synoptic scales.

RWPs are the main driver of extratropical weather and climate, also at larger spatial and time scales. Variations of the zonal-mean westerlies of the order of 2-3 weeks were found to originate from energy conversion between zonal mean flow and medium scale waves in the SH (Webster and Keller, 1975; Randel and Stanford, 1985). Eddy feedbacks are responsible for the increased time scale of annular modes in the extratropical winter troposphere which are coupled with the stratospheric polar vortex behaviour (Lorenz and Hartmann, 2001, 2003), making RWPs an important contributor to stratosphere-troposphere
coupling (Kidston et al., 2015).

The theory for lower-upper level PV anomaly co-amplification (Hoskins et al., 1985), idealized life-cycles (Gall, 1976; Simmons and Hoskins, 1978; Thorncroft et al., 1993), conceptual and case studies (Shapiro and Keyser, 1990), all have highlighted the importance of tropopause processes in the formation and evolution of baroclinic waves for decades, but only recently research has started to focus on the role of the stratosphere. Williams and Colucci (2010) showed that RWP properties in the
upper troposphere depended on lower stratospheric conditions, namely the strength of the polar vortex. New case studies about extreme cyclones are pointing out to a significant role of stratospheric conditions in their development (Odell et al., 2013; Tao et al., 2017a, b), and a more general study about severe European windstorms found that in 20 of 60 cases the stratospheric contribution during their deepening phases was over 10% (Pirret et al., 2017). Nevertheless, despite their interaction with the



stratosphere, it is generally assumed that RWPs cannot propagate upward due to the typical wind regimes in the stratosphere (Charney and Drazin, 1961).

There are several issues concerning RWP representation in forecast models. Atmospheric models present a general bias in their simulated waveguides, which are not as sharp as observed (Gray et al., 2014; Giannakaki and Martius, 2016). The hori-
zontal and vertical structures of forecast errors tend to spread together with the RWP's envelope (Dirren et al., 2003; Hakim, 2005; Sellwood et al., 2008; Zheng et al., 2013), and particularly near-tropopause dynamics are the ones dominating error growth (Baumgart et al., 2018). Apart from baroclinic energy generation, other important processes that affect PV distribution in RWPs near the tropopause (and thereby forecast error growth) are latent heat release and longwave radiative cooling (e.g., Martínez-Alvarado et al., 2016; Teubler and Riemer, 2016). The organized structure of RWPs and their large travelling dis-
tances, compared to individual troughs/ridges or cyclones/anticyclones, offer the opportunity of improving and extending the range of weather forecasts with a better understanding and modelling of their dynamics (Grazzini and Vitart, 2015).

Climate change can have opposing influences on the storm tracks (Shaw et al., 2016) or the polar front jet (Hall et al., 2015) position. As an example, Arctic amplification will decrease the lower troposphere's meridional temperature gradient and baroclinicity, tending to shift the storm track equatorward; while stratospheric radiative cooling by $CO_2$ will increase
near-tropopause baroclinicity towards the pole, counterbalancing the shift in the lower troposphere. This is one reason why climate projections show sensitivity to the inclusion of the stratosphere in models (Scaife et al., 2012), and RWP dynamics play a central role in establishing where the projected storm tracks and polar front jets will shift to. Models used for climate projections, e.g. the Coupled Model Intercomparison Project Phase 5 (CMIP5) are known to have biases in their storm tracks (Chang et al., 2012; Hall et al., 2015).

Recognizing the importance of RWPs for weather forecasting, climate change projections and stratosphere-troposphere interactions, the scientific community would greatly benefit from increased observational knowledge of RWP structures and behaviour across the upper-troposphere and lower-stratosphere (UTLS), which is the main goal of our study in an area lacking research thus far. The recent availability of Global Navigation Satellite System radio occultation (GNSS-RO) measurements (e.g., Kursinski et al., 1997) enables the study of RWP temperature and pressure anomalies with global coverage across the
UTLS at unprecedented high vertical resolution. The exclusive use of one decade of GNSS-RO observations (2007-2016) will avoid any limitation from reanalysis or model output in the form of biases or lack of vertical resolution within the UTLS. The present manuscript intends to describe general and purely observational RWP properties across the UTLS.

GNSS-RO observations have been used to study atmospheric waves in many recent studies. In the equatorial regions, Kelvin waves have been studied the most so far (e.g., Randel and Wu, 2005; Flannaghan and Fueglistaler, 2013; Scherllin-Pirscher
et al., 2017), with Zeng et al. (2012) and Kim and Son (2012) also studying the Madden-Julian Oscillation (MJO, Madden and Julian (1994)). Alexander et al. (2008) and Pilch Kedzierski et al. (2016a) had a wider approach and considered all equatorial wave types. In the extratropics, studies using GNSS-RO measurements have concentrated mainly on the extraction of gravity wave parameters (e.g., Tsuda et al., 2000; Wang and Alexander, 2010; Kohma and Sato, 2011; Tsuda, 2014; Schmidt et al., 2016). Madhavi et al. (2015) studied the two-day wave in the upper stratosphere and lower mesosphere in both the
NH and SH extratropics. Alexander and Shepherd (2010) studied planetary wave activity in the Arctic and Antarctic lower





and mid-stratosphere; while Shepherd and Tsuda (2008) studied planetary waves in the SH polar summer at 30 km height. Pilch Kedzierski et al. (2017) used GNSS-RO to study extratropical tropopause modulation by waves, although only separating wave activity by time-scale and propagation direction, without describing wave properties. Our study is a first attempt to describe RWP properties in the extratropics with the use of GNSS-RO.

This paper is organized as follows: section 2 will introduce the GNSS-RO missions used for our study and the analysis methods applied, section 3 will present how the zonally averaged RWP activity is distributed in the UTLS globally, and section 4 will concentrate on RWP zonal and vertical structures. Both sections 3 and 4 will start with case studies to introduce how RWPs evolve across the UTLS, moving on to climatological statistics with a focus on general and common RWP properties in the extratropics of both hemispheres. Section 5 will discuss some implications of our results, and section 6 will summarize our
main findings.

## 2 Data and methods

### 2.1 Datasets

We make use of global navigation satellite system radio occultation (GNSS-RO) measurements from the following satellite missions: CHAMP (Wickert et al., 2001), COSMIC (Anthes et al., 2008), GRACE (Beyerle et al., 2005), Metop-A (von
Engeln et al., 2011), the successive Metop-B, SAC-C (Hajj et al., 2004), and TerraSAR-X (Beyerle et al., 2011). All data are re-processed or post-processed occultation profiles with moisture information ('wetPrf' product) from the COSMIC Data Analysis and Archive Center (CDAAC, https://cdaac-www.cosmic.ucar.edu/cdaac/products.html), for the years 2007-2016. GNSS-RO profiles of temperature and pressure are provided interpolated on a 100 m vertical grid between the surface and 40 km height, although their effective physical resolution varies from ∼1 km in regions of constant stratification down to the order
of 100 m where stratification gradients occur, such as the tropopause or the top of the boundary layer (Kursinski et al., 1997; Gorbunov et al., 2004).

Different studies have shown the consistency, mission-independence and good precision among different GNSS-RO satellite missions as well as compared to radiosondes (Hajj et al., 2004; Wickert et al., 2009; Schreiner et al., 2011; Anthes et al., 2008; Ho et al., 2017). The advantage of GNSS-RO profiles over radiosondes relies on their global coverage, weather-independence and higher sampling density. The time period of our analysis spans 2007-2016, and thanks to the simultaneous use of several
missions the total number of GNSS-RO profiles is rather stable with around 2500 profiles per day.

All GNSS-RO profiles undergo an in-depth quality check (see section 2.2) to eliminate erroneous/unphysical profiles which, if present often enough, can trigger false signals when applying space-time filters. After merging all GNSS-RO data into one gridded dataset (see section 2.3), we find no discontinuities or artefacts arising from the presence of GNSS-RO from different
missions, due to the self-calibration and consistency of each satellite instrument and the post(re)-processing at the same centre (CDAAC).

The GNSS-RO measurements are complemented with zonal mean zonal wind data from the ERA-Interim reanalysis (Dee et al., 2011) for the same period 2007-2016, in order to provide a context about background wind regimes. The programming





language 'R' (R Core Team, 2015) is used to perform the quality control (section 2.2), gridding (section 2.3) and wave analysis (section 2.4) from the GNSS-RO data.

## 2.2 Quality control for GNSS-RO profiles

The quality control for the GNSS-RO measurements from all satellite missions is performed in two steps:

1. The first step is intended as a general screening for profiles whose temperatures or tropopause stability fall outside global climatological values. Temperature profiles with values < -100°C or > 50°C are excluded, as well as those with T < -90°C above 35 km height, since the coldest values within the winter polar vortex occur well below 35 km. The tropopause height ($TP_z$) is defined following the World Meteorological Organization lapse-rate criterion (WMO, 1957). Profiles where the tropopause cannot be found, or those with tropopauses unreasonably warm (> -45°C), are excluded. This is close to the mean tropopause temperature in polar summer (-50°C) but we do not find discontinuities in the time availability of GNSS-RO profiles arising from this criterion. Static stability is calculated as the Brunt-Väisälä frequency squared, $N^2 = (g/\Theta)\cdot(\partial\Theta/\partial z)$, where g is the gravitational acceleration, $\Theta$ the potential temperature, and $\partial z$ its vertical derivative. The $N^2$ maximum above $TP_z$ (up to 3 km) is computed, and profiles with tropospheric ($N^2 < 3\times10^{-4}s^{-2}$) or too high values ($N^2 > 100\times10^{-4}s^{-2}$) are excluded. These conditions are based on Son et al. (2011) for $TP_z$ temperature climatologies from GNSS-RO measurements, and Pilch Kedzierski et al. (2016a) for maximum $N^2$ climatologies, whose highest values are found in the tropics. From the initial 10,053,153 profiles available for 2007-2016 from all GNSS-RO satellite missions, this first step keeps 9,369,092 (93.2%) of them.

2. After step 1 there remains a significant number of GNSS-RO profiles whose stratospheric temperatures markedly stand out when compared to nearby occultations in space and time, although passing the climatological criteria. On a global 5° by 5° longitude-latitude grid, for each day throughout 2007-2016 and at every grid point, GNSS-RO profiles within +-3 days, +-10° longitude and +-5° latitude are selected. A distribution of their 30 km temperature is computed, and a mean temperature profile is calculated from those GNSS-RO profiles that fall between the 0.2 and 0.8 quantiles, which avoids the influence of possibly erroneous profiles. The integrated squared temperature difference from the mean profile between 20 and 40 km height is calculated for each occultation: $A_i = \sum_{20km}^{40km}(T'_i)^2$. The 0.5 quantile of A represents the half of the selected GNSS-RO profiles that are closer to the mean profile. The profiles with $A_i$ exceeding 20 times the 0.5 quantile (that fall far out of the distribution) are excluded. Step 2 eliminates GNSS-RO profiles with unrealistic stratospheric temperatures for their season and location, and even in extreme situations with stark temperature contrasts such as sudden stratospheric warmings this criterion is not met.

After the quality control is carried out 9,215,804 profiles (91.7% of the initial GNSS-RO dataset) are used.

## 2.3 Gridding

The method to grid GNSS-RO profiles in our study is a refinement of Pilch Kedzierski et al. (2017) and Pilch Kedzierski et al. (2016a), with better horizontal resolution and with ground-based instead of tropopause-based averaging. Both were developed





after the method by Randel and Wu (2005). GNSS-RO profiles of temperature and pressure for the years 2007-2016 are gridded daily on a 5° by 5° longitude-latitude grid, from 85°S to 85°N. The height range analysed is between 6 km and 40 km, although because the focus of this manuscript is the UTLS region most of the presented results will use a lower lid. The amount of GNSS-RO profiles that penetrate deeper than 6 km diminishes at lower altitudes. Less data available to grid forces more
interpolation and less reliability of filtered signals at lower levels. The GNSS-RO profiles that fall within the grid point area are averaged following:

$$T_{grid}(\lambda, \phi, z, t) = \sum_i w_i T_i(\lambda, \phi, z, t) / \sum_i w_i \tag{1}$$
$$P_{grid}(\lambda, \phi, z, t) = \sum_i w_i P_i(\lambda, \phi, z, t) / \sum_i w_i \tag{2}$$

where $\lambda$ is longitude, $\phi$ is latitude, z is height and t is time. The weight $w_i$ is a Gaussian function determined by each
GNSS-RO profile's distance from the grid centre:

$$w_i = exp(-[(D_x/2.5)^2 + (D_y/2.5)^2 + (D_t/12)^2]) \tag{3}$$

with $D_x$ and $D_y$ as the distances in °longitude and °latitude from the grid's centre, and $D_t$ as the time distance in hours from 12UTC, divided by the grid's half-size in all dimensions: 2.5° longitude, 2.5° latitude and 12 hours.

Typically 2-3 GNSS-RO profiles of the same day are selected within the grid's area for averaging, with the following
exceptions. In a 28% of grid points the allowed distance to search for GNSS-RO profiles needs to be expanded to +-5° longitude and +-3.5° latitude in order to avoid gaps (in 9% of cases it expands further to +-10° longitude and +-5° latitude), without changing the weighting function $w_i$ in any case. The remaining 1% of gaps are filled by averaging adjacent grid points (+-1 in longitude, then +-1 in time). The exception's percentages presented here belong to the Northern hemisphere from 15° to 85° latitude and are nearly identical in the Southern hemisphere. The gridded tropopause height $TP_z(\lambda, \phi, t)$ is calculated with the
same weighting of all profiles' tropopauses.

Overall the real resolution of the gridded dataset is slightly coarser than 5° longitude-latitude and may be viewed as an interpolation to a certain degree. Nonetheless this setting resolves the horizontal scale of the RWPs analysed in our study very well (see sections 3 and 4).

### 2.4 Wave analysis

Given that the main goal of this paper is to describe properties of RWPs in the UTLS in a very general manner, we intend to keep the wave analysis as simple as possible. In most analyses a Fast Fourier Transform (FFT) in longitude is used to extract the wave anomalies from the gridded GNSS-RO temperature and pressure data, either for individual wavenumbers (section 3) or using an intermediate range typical of RWPs (section 4). For RWP envelope reconstruction, the Hilbert transform (Zimin et al., 2003) is applied on RWP pressure anomalies without any further pre- or post-processing of the RWP anomalies or their
envelope (section 4). A couple of exceptions that add a degree of complexity to the analysis are disclosed below.





**FFT in longitude and time**: To avoid including stationary waves when showing the climatological distribution of RWP activity over latitudes, we use a two-dimensional FFT in longitude and time (Schreck, 2009) to keep the transient components of each wavenumber (eastward-propagating, 2-20 day periods). This is only used for Figures 3 and 4 in section 3.2.

**Vertical scale analysis**: RWP anomalies are extracted by taking zonal wavenumbers 4-8 with a FFT in the longitude dimen-
sion, obtaining daily longitude-height snapshots. In a second step, at every longitude grid an additional FFT is performed in the vertical dimension on the profile of RWP anomalies between 6-36 km height, in order to obtain the power spectrum of the different vertical wavelengths. The power spectra are then averaged for mid-latitudes in Fig. 10 in section 4.2.

**Zonality of RWP envelopes**: A FFT in longitude is used to obtain RWP anomalies (w4-8), on which the Hilbert transform is applied to reconstruct the RWP envelope. In a third step, another FFT in longitude is applied on the reconstructed envelope to
obtain its zonal wavenumber power spectrum. The average wavenumber spectra of RWP envelopes at mid-latitudes are shown in Fig. 12 in section 4.2.

## 3   Distribution of wave activity in the extratropical UTLS

Section 3 concentrates on the analysis of wave activity as the amplitude of individual harmonics, which only allows to study RWPs from a zonally integrated perspective. Subsection 3.1 will introduce examples of wave activity behaviour from gridded
temperature and pressure GNSS-RO data for one year and one latitude band: July 2008 till June 2009 at 50°N. Subsection 3.2 will generalize those results with climatological statistics for the 2007-2016 period throughout the whole extratropics. Results are presented this way so the reader gets an overall impression of where wave activity is located as well as how it evolves on a day-to-day basis.

Section 4 will add upon the latter by analysing RWP zonal structures and their horizontal propagation, which appear from
the combination of a range of intermediate wavenumbers that shape the characteristic carrier wave and envelope of RWPs (idealized examples are nicely formulated in Zimin et al. (2003) and Wolf and Wirth (2015)).

### 3.1   Time-height section examples

We begin by analysing RWP activity in time-height sections. Figure 1 shows the evolution of the amplitudes of wavenumbers 4 to 8 during 2008/2009 at 50°N, filtered from gridded GNSS-RO temperature fields. The 50°N latitude represents the one
with most wave activity in the NH extratropics. The highest amplitudes (orange and red shading) are found directly above the zonal mean lapse-rate tropopause (magenta line) and in the upper troposphere, with a stark minimum in temperature signals in between (blue shading). This vertical distribution of wave activity in terms of temperature tightly follows the zonal mean tropopause over time. Wavenumbers 4 and 5 reach amplitudes of 6-7 K, with higher wavenumbers showing lower amplitudes: 3-4 K for w8.

The activity of all wavenumbers in general seems to be confined within westerly winds (solid grey lines): very little penetrates into the summer easterlies (dashed white lines). Interestingly, stratospheric zonal mean zonal winds of 10-20 m/s are no impediment for wave activity to propagate beyond 20 km height, which can be seen most clearly for wavenumbers 4 to 6: they



show amplitudes in excess of 4 K around 20-26 km height before and during the 2009 sudden stratospheric warming (SSW, see the patch of easterly winds starting in February 2009 at 26 km). All wavenumbers show temperature signals of 2-3 K (yellow shading) reaching 18-20 km height very often. The observed GNSS-RO temperature signals of wavenumbers 4-8 reaching that high into the stratosphere in Fig. 1 is in contradiction with the Charney-Drazin criterion (Charney and Drazin, 1961).

In Figure 1 it can also be noticed that the temperature signals in the lower stratosphere (LS) and the upper troposphere (UT) tend to amplify and dissipate simultaneously, apart from showing very similar amplitudes. For wavenumbers 6 and higher this happens throughout the whole year; for w5 and w4 less during the summer months. This indicates the existence of co-amplification between the UT and LS temperature signals, in addition to the co-amplification of potential vorticity (PV) anomalies between the surface and UT (Hoskins et al., 1985). The vertical propagation of the temperature signals is difficult
to discern in Fig. 1. There are hints of rapid upward propagation in the stratosphere which are better visible in the analysis of pressure signals, which will be discussed later in this section.

It shall be pointed out that the wave signals shown in Fig. 1 mostly belong to travelling waves: if the same Figure is done with signals filtered also in time (eastward-propagating with 2-20 day periods), it looks very similar (not shown). We proceed to repeat the analysis from Fig. 1 on GNSS-RO gridded pressure fields. The filtered pressure anomalies are displayed relative
to the mean pressure of their vertical level (in % units) to make the comparison between tropospheric and stratospheric signals more fair, given the exponential decrease of pressure with height.

Figure 2 shows time-height sections of the pressure amplitudes (in % of the zonal mean pressure) of wavenumbers 4 to 8, also for 50°N and 2008/2009. The same bursts of wave activity can be observed as in Fig. 1, but the most striking difference is that the pressure anomalies in Fig. 2 maximize around tropopause height, following it constantly throughout the year for
all studied wavenumbers. Wavenumbers 4 and 5 reach amplitudes of 3-4% (dark red and brown shading), diminishing towards higher wavenumbers: ∼1.5% for w8.



**Figure 1.** Time-height sections of wave activity for wavenumbers 4-8 at 50°N, in terms of the amplitude of the individual harmonics in temperature (colours). Magenta line denotes the zonal-mean $TP_z$. Grey solid lines denote westerly zonal winds, with 10 ms$^{-1}$ separation. Thick white solid and dashed lines denote 0 ms$^{-1}$ and -3 ms$^{-1}$, respectively. To improve visibility, the ERA-Interim zonal-mean zonal wind is displayed with a running mean of +-15 days.





**Figure 2.** Same as in Fig. 1 but for wave pressure anomalies, relative to the mean pressure of each vertical level.





In Fig. 2 the pressure anomalies appear to form around tropopause height and radiate outward vertically, diminishing their amplitude away from the tropopause. Wavenumbers 4 and 5 (and w6 in smaller amounts) show frequent upward propagation in the winter stratosphere well beyond 20 km height. Wave activity in terms of pressure of w7 and w8 tends to be confined closer to the tropopause, with very little activity reaching beyond 18 km height in Fig. 2. Compared to Fig. 1, the pressure

anomalies from Fig. 2 show a much better-defined continuity in their upward propagation in the stratosphere, making them much easier to track in time-height sections. For example, w4 and w5 temperature anomalies in January 2009 in Fig. 1 show no attachment between the lower and middle stratosphere (red patches are separated by blue regions of very low temperature amplitudes). Meanwhile, the corresponding pressure anomalies in Fig. 2 show clear continuity and upward propagation from the lower to the middle stratosphere. This can be discerned in the w4 bursts in the beginning of December 2008, mid-March

2009 and mid-April 2009, with the red and yellow tracks being tilted upward in the positive time direction in Fig. 2. In January 2009, vertical propagation of several w5 bursts up to 26 km height is observed coinciding with the 2009 SSW. In this case, w4 vertical propagation seems to be very rapid in January 2009 (the red track appears quasi-vertical), maintaining very high wave amplitude throughout the whole lower and middle stratosphere. Interestingly w4, w5 and w6 to some degree show some re-amplification of their pressure anomalies near the zero-wind line (thick solid white line in Fig. 2), which is propagating

downward during the 2009 SSW. This should not be interpreted as downward propagation of the RWP pressure anomalies, only that they encounter their critical level for upward propagation there, amplifying before breaking.

Overall the temperature anomaly structures in Fig. 1 appear much noisier and with more discontinuities compared to the same analysis done with pressure data in Fig. 2, which provides a much cleaner view of wave activity location and propagation. One explanation for this could be that the pressure data show the dynamical structure of the wave which remains together as

one system. The thermal structure of the same wave is subject to more variations due to heat fluxes, meridional temperature gradients, energy conversion, etc., which feed the wave but need not be proportional to its dynamical structure locally. To our knowledge, our study is the first to analyse wave activity in terms of pressure with GNSS-RO data: only temperature or parameters derived from it have been used to study waves thus far. From the results presented in Fig. 2, the benefits of analysing pressure GNSS-RO data are noticeable.

From Figs. 1 and 2 it can be concluded that the separate temperature anomalies in the UT and LS and their coamplification are part of the same pressure system that goes across the UTLS and occasionally propagates far up into the stratosphere, more often for wavenumbers 4 and 5. The pressure anomalies in Fig. 2 that maximize around the zonal mean tropopause height can be considered equivalent to the upper-level PV anomalies (Hoskins et al., 1985) which then couple with surface circulation. Our results are in agreement with Teubler and Riemer (2016) who showed PV anomalies maximizing around tropopause

height. Since our gridded GNSS-RO dataset includes altitudes of 6 km and higher, the wave behaviour in the lower troposphere cannot be diagnosed to study its coupling with the near-tropopause anomalies, but this will be the subject of future work in combination with other datasets.

Another topic of high interest for future work is the behaviour of RWP activity, namely w4 and w5, and the possible influence on SSWs since in Fig. 2 we observe increased propagation into the stratosphere around the time of the 2009 SSW. We see a

similar behaviour of RWP activity in the 2010 and 2013 SSW cases (see and compare Figs. S1, S2 and S3 in the supplement).





Domeisen et al. (2018) showed important phase speed and amplitude changes of w1 and w2 at the 100 hPa level (∼16 km height) preceding the onset of SSWs. The lower stratosphere plays a key role in controlling the downward propagation of SSW anomalies and their coupling with the troposphere: Karpechko et al. (2017) showed that Northern annular mode anomalies at the 150 hPa level define which SSWs affect the troposphere. RWP activity is known to force low-frequency variability of the

zonal mean circulation (Webster and Keller, 1975; Randel and Stanford, 1985).

Given the large amounts of RWP activity we diagnosed in the LS (and stratosphere during SSWs, see Fig. 2), RWP interaction with the LS mean flow through which planetary waves propagate should be expected: this would affect LS NAM conditions that control SSW downward propagation (Karpechko et al., 2017), as well as indirectly affect the phase speeds of upward-propagating planetary waves (Domeisen et al., 2018) by Doppler-shifting. One may even speculate RWPs could

*directly* affect planetary wave phase speeds and amplitudes in the LS if some wave-wave interaction and energy transfer or positive interference would happen between the two. In either case, we state that RWP activity in the LS is potentially relevant for both onset and downward propagation stages of SSWs, opening an interesting field of research.

Figures 1 and 2 have shown an example for one year (July 2008 till June 2009) and one latitude band: 50°N. Section 3.2 below will generalize the results shown above, giving climatological statistics for all extratropical latitudes from a decade

(2007-2016) of GNSS-RO data.

### 3.2   Climatological distribution of wave activity over latitude and height

How is the wave activity of individual wavenumbers distributed over the whole extratropical latitude range? Taking the zonal mean tropopause level as reference, since wave activity in terms of pressure follows it and maximizes there (Fig. 2), we calculate the mean amplitude of each wavenumber's filtered pressure signal. We select the eastward-propagating components with 2-20

day periods, in order to isolate travelling waves throughout the NH extratropics. The resulting wave spectrum is shown in Fig. 3a for the whole NH gridded GNSS-RO dataset (2007-2016), and separated into winter and summer climatologies in Fig. 3 (b-c).

The spectrum from Fig. 3a has a similar shape to the spectrum from Wolf and Wirth (2017) (their Fig. 6). Although the parameter used by Wolf and Wirth (2017) is 300 hPa meridional winds and the time filtering is slightly different (30 day high-

pass), their spectrum also shows the two relative maxima of wave activity: one at mid-latitudes around w6 and the other at polar latitudes and lower wavenumbers, similarly to our Fig. 3a. Travelling waves tend to have a similar range of wavelengths at all latitudes, and the spectrum shape results from their resulting zonal wavenumber Fourier composition at the different latitudes.

The wave spectrum for NH winter (Fig. 3b) is more elongated towards lower latitudes, reaching 30°N. Meanwhile in the NH summer spectrum (Fig. 3c) there is very little wave activity south of 45°N, which agrees well with the seasonality of the

jet stream position. Interestingly, transient wave activity in the Arctic region (poleward of 70°N, w1-3) seems to be very active all year round at the zonal mean tropopause level.

We take the timeseries of the amplitude of wave temperature anomalies 1.5 km above and 3 km below the zonal mean tropopause at each latitude band and separate wavenumber as a measure of LS and UT thermal wave activity. These heights were chosen to avoid the minimum in wave activity in terms of temperature around and right below the tropopause in Fig. 1.



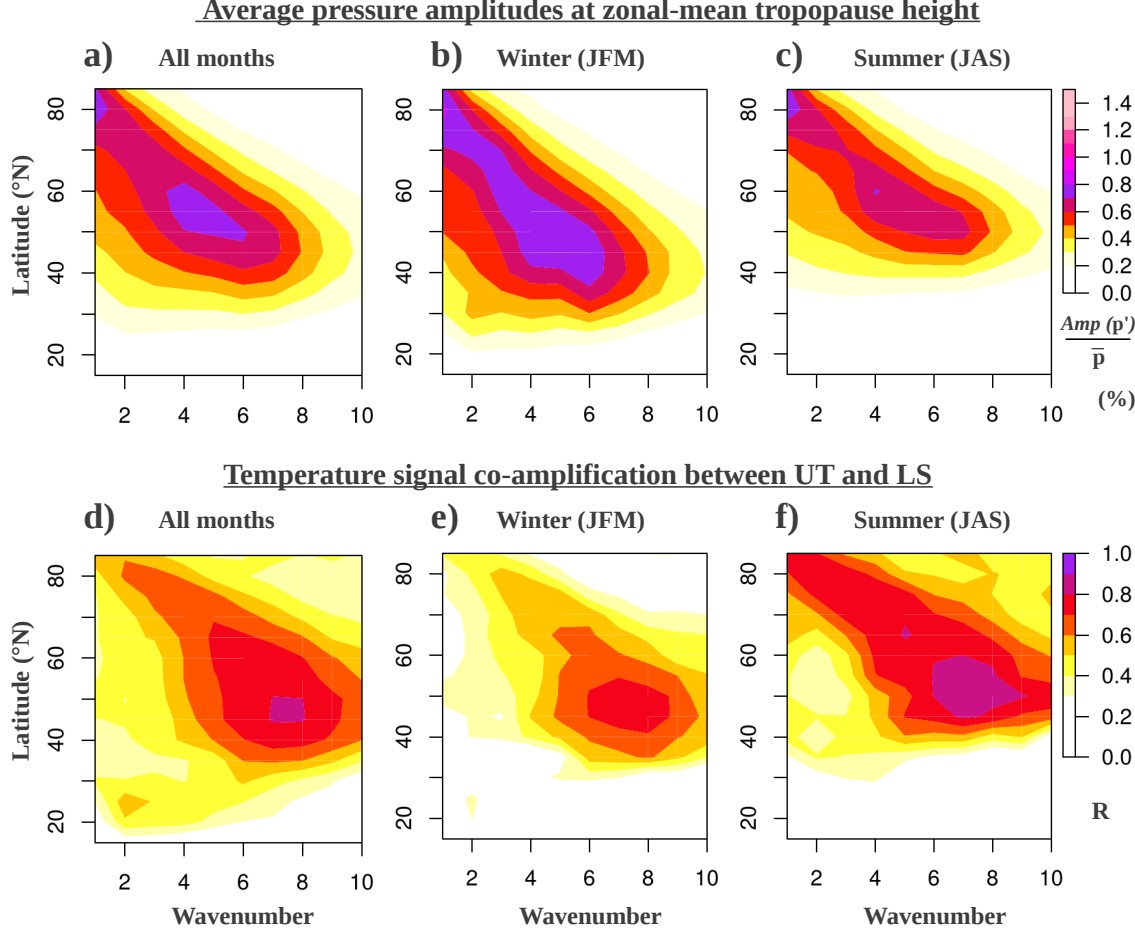

**Figure 3.** Upper row: wave activity spectra, as the mean amplitude of the filtered pressure anomalies at the zonal-mean tropopause level for (a) the whole 2007-2016 series, (b) winter and (c) summer months. Bottom row: co-amplification of the LS and UT temperature signals, as the correlation of their amplitude time-series for (d) all, (e) winter and (f) summer months.

The correlation of the LS and UT timeseries will indicate the degree of their co-amplification, which is shown in Figures 3 (d-f). After seeing Figs. 1 and 2, one would expect a tendency of the upper and lower rows in Fig. 3 to match, but this is not the case: higher values of the measure for co-amplification are shifted towards higher wavenumbers at all seasons. Furthermore, the NH summer (Fig. 3f) shows markedly higher correlations than the NH winter season (Fig. 3e). The results shown in Figure 3 (d-f)

5    indicate that the co-amplification of wave temperature signals between the UT and LS is done more towards the synoptic-scale and the upper end of the medium-scale of the wave spectrum, at any season. This is an unexpected finding. One possibility to explain it would be that the temperature signal requires meridional advection to be formed, and meridional winds have been shown to have a trough-ridge asymmetry (Wolf and Wirth, 2015, 2017) due to their semi-geostrophic nature, with strongest meridional winds found closer to the trough's centre. Therefore meridional winds and temperature could typically have slightly

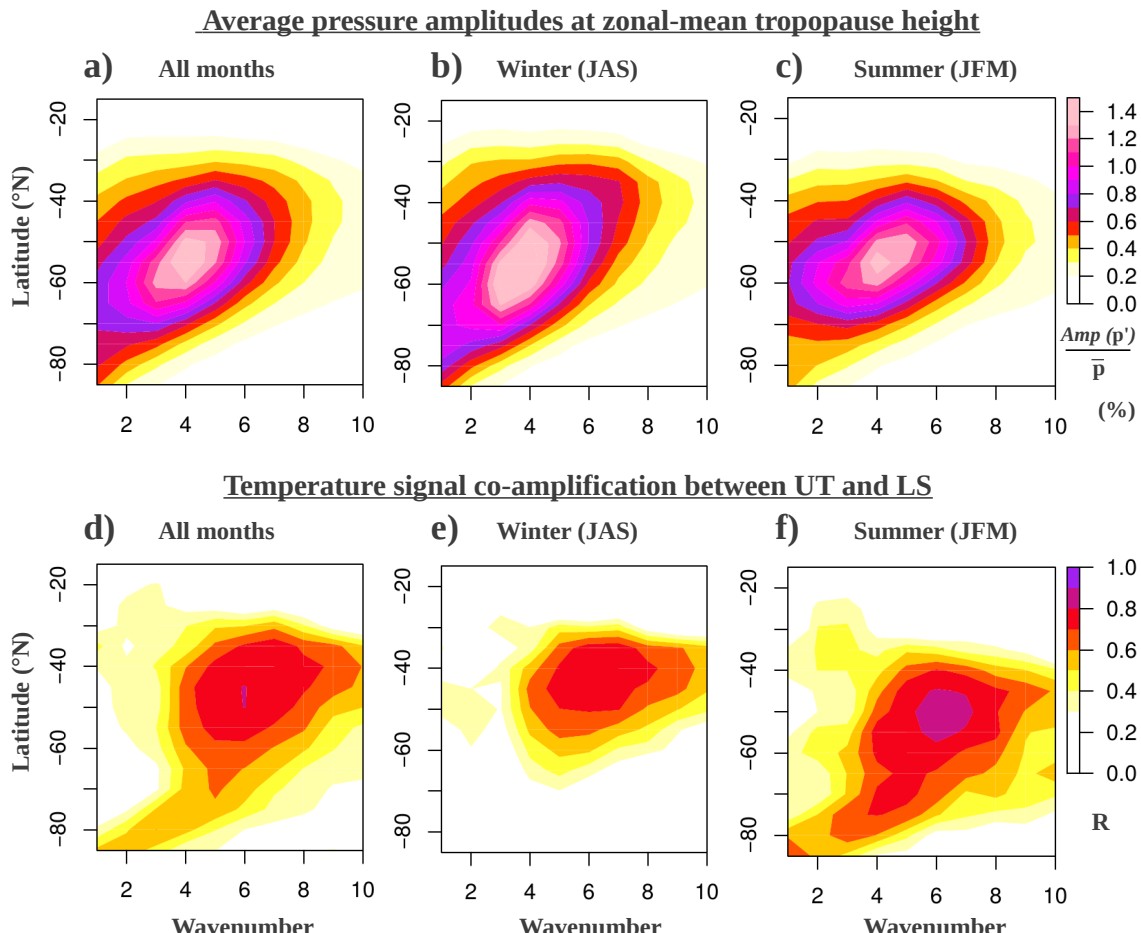

**Figure 4.** Same as Fig. 3 but for the Southern hemisphere.

shorter wavelengths compared to their corresponding pressure anomalies' wavelength. Proving this would require the use of other datasets beyond GNSS-RO, thus this is beyond the scope of our study.

We repeat the same analysis from Fig. 3 with RWP activity latitudinal distribution and the UT-LS co-amplification measure in the Southern hemisphere (Figure 4). The wave spectra, as well as the UT-LS co-amplification measure, are qualitatively

5 similar in the NH (Fig. 3) and SH (Fig. 4). The SH shows maximized wave activity at w4 and ~55°S throughout the year, with markedly higher values in terms of mean relative pressure amplitude: SH maxima reach ~1.3% in Fig. 4 (a-c), versus ~0.7% in the NH in Fig. 3 (a-c). SH mid-latitudes show increased eastward-propagating wave activity overall, maximized at lower wavenumbers compared to the NH mid-latitudes, in agreement with previous studies (see Randel and Stanford (1985) and references therein). This interhemispheric difference in the wave spectrum can be explained by the absence of mountains

10 in the SH: a model study by Hayashi and Golder (1983) noted a similar change in the wave spectrum when NH mountains were removed.



**RWP activity at 40-60° latitude**

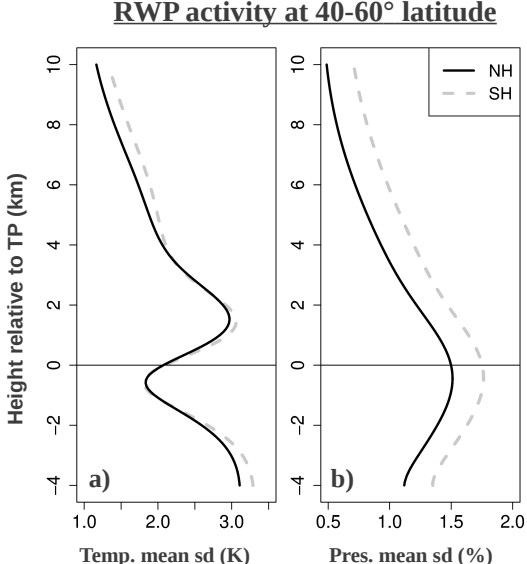

**Figure 5.** Vertical profiles of (a) temperature and (b) pressure time-mean RWP activity within 40°-60° latitude, calculated as the mean standard deviation of w4-8 daily anomalies and relative to the zonal-mean tropopause height.

The maximum of SH transient wave activity shows little seasonal variation in the latitude where it is located (Fig. 4 b and c), although in winter it shows a broader latitudinal extent, indicating larger meridional advection scales. In contrast to the NH, the SH polar latitudes show very little wave activity during summer months (Fig. 4c). As in the NH, the co-amplification of SH wave temperature signals in the UT and LS is maximized at higher wavenumbers and during the summer months (Fig. 4 d-f).

The SH distribution of the UT-LS correlation factor, relative to the distribution of transient wave activity, is very similar to the NH both in magnitude and shape.

Figures 3 and 4 summarized the distribution of transient wave activity over latitudes and wavenumbers in the NH and SH extratropics. We proceed to show a climatology of the vertical distribution of RWP activity in both hemispheres. We noted earlier in Figs. 1 and 2 that the temperature signal had a relative minimum near and right below the zonal mean tropopause, and

a maximum located right above it; while the pressure signal maximized and followed the zonal-mean tropopause. We take the standard deviation (std) of zonal wavenumbers 4-8 daily temperature and pressure anomalies, and average the daily std value relative to the zonal mean tropopause level at both NH and SH mid-latitudes. We select w8 as the upper limit because we note very little wave activity at higher wavenumbers from Figs. 3 and 4. The resulting mean vertical profiles of RWP activity are shown in Figure 5.

The vertical distribution of the RWP temperature signals in Fig. 5a is very similar in both NH and SH mid-latitudes, with tropospheric and LS relative maxima above ∼3 K, and a minimum close to and below the zonal mean tropopause level of ∼1.5 K. The pressure signals in Fig. 5b maximize around and below zonal mean tropopause height, with the SH reaching slightly higher values (∼1.8%) than the NH (∼1.5%), but otherwise alike in shape. The higher RWP activity in the SH was already noted





comparing the wave spectra from Figs. 3 (a-c) and 4 (a-c). The climatology in Fig. 5 shows the expected vertical distributions from the 50°N examples from Figs. 1 and 2, and shows very little interhemispheric differences. Our climatological profile of RWP temperature signals (w4-8) qualitatively agrees very well with the seasonal averages for individual wavenumbers (w4-7) by Randel and Stanford (1985), their Fig. 4; and also with the vertical distribution of sensible heat fluxes in Blackmon and White (1982), their Figs. 4-5. Our climatological profile of RWP pressure signals shows very good agreement with the vertical distribution of momentum fluxes in Blackmon and White (1982), their Figs. 7-8; as well as the seasonal vertical profiles of w5 geopotential height amplitudes in the SH in Hirooka et al. (1988), their Fig. 8a. Also, early model experiments predicted RWP activity in terms of geopotential or eddy kinetic energy to maximize around the tropopause (Gall, 1976; Simmons and Hoskins, 1978).

## Short summary

This section introduced the latitudinal, hemispheric, vertical and wavenumber distribution of RWP activity obtained using gridded temperature and pressure GNSS-RO observations exclusively. Our results from a zonally averaged perspective agree very well with early reanalysis studies (e.g., Blackmon and White, 1982; Randel and Stanford, 1985) as well as early idealized model experiments (Gall, 1976; Simmons and Hoskins, 1978; Hayashi and Golder, 1983). We summarize the main new findings of this section:

1. RWP activity is centered around and follows the zonal-mean tropopause level during all seasons (Figs. 1, 2 and 5). Our results may have implications for improving existing RWP diagnostics: the choice of a specific vertical level could have aliasing effects on the obtained RWP amplitudes (see discussion in section 5).

2. Pressure anomalies are much more reliable to track the vertical propagation of a RWP than temperature ones. We noted episodes of high amplitude RWP activity propagating into the NH middle stratosphere, especially wavenumbers 4 and 5. In Fig. 2 this coincided with the 2009 SSW, and we note similar RWP vertical propagation in the 2010 and 2013 SSW cases (see Figs. S1, S2 and S3).

3. The co-amplification of RWP temperature signals in the UT and LS is more effective towards synoptic-scale and the higher end of medium-scale wavelengths, as shown by the higher correlation factors being shifted towards higher wavenumbers in Figs. 3 (d-f) and 4 (d-f).

## 4   Zonal and vertical structures of RWPs

After analysing RWP activity from a zonally averaged perspective in Section 3, this section will focus on the RWP zonal structures. A classical RWP example with eastward phase propagation and faster group speed will be presented in section 4.1 with the use of Hovmöller diagrams and longitude-height snapshots of RWP anomalies from the gridded GNSS-RO data.

Recent studies have pointed out that under specific resonance conditions, RWPs can become stationary and form large-scale teleconnection patterns in mid-latitudes, which have been linked to the extreme heatwaves of 2003, 2010 and 2018 (Petoukhov





et al., 2013; Kornhuber et al., 2017, 2019). To test whether RWP properties in the UTLS are different under these conditions, we will repeat the same analysis from section 4.1 during the Moscow heatwave in summer 2010, which will be presented in section 4.2. After these two case-studies, in section 4.3 we will present climatological statistics of the vertical scale of RWP anomalies, the relation of UT and LS zonal temperature structures, and the zonal scale of RWP envelopes for both NH and SH.

## 4.1   Classical RWP example

RWP activity was shown to follow the zonal mean tropopause in Figs. 1 and 2, therefore we take this level to extract RWP pressure anomalies from the gridded GNSS-RO data. This case study was selected to show a classical RWP with eastward phase propagation and a high-amplitude envelope of faster speed, which is the case at the end of February and beginning of March 2009 at 40-60°N. We select pressure anomalies belonging to wavenumbers 5 to 8 in order to avoid stationary waves with w1-4 present at the time. The evolution of the RWP pressure anomalies is shown in the Hovmöller diagram in Figure 6a, and the RWP envelope after applying the Hilbert transform (Zimin et al., 2003) to the w5-8 anomalies is shown in Fig. 6b.

In Fig. 6b the formation of a RWP around 2009-02-27 over the Pacific Ocean ($\sim$150°W) can be observed, with the envelope propagating eastward until 2009-03-07 when it covers the Atlantic Ocean (60-0°W). The eastward phase propagation of the RWP (individual blue troughs and red ridges) in Fig. 6a can be compared to the markedly faster movement of the envelope (green and dark colors) in Fig. 6b. This RWP reaches amplitudes exceeding 4‰. By 2009-03-07, a new RWP is forming again over the Pacific Ocean, this one with an even faster envelope and circumnavigating the globe by 2009-03-12, and even reaching global scale for a couple of days although having lower amplitudes than the first RWP, of around 2-3‰. It also has to be noted that low-amplitude fluctuations of $\sim$1‰ within the w5-8 range are present almost constantly (e.g., light yellow shading in Fig. 6b).

Recent studies using meridional winds and the Hilbert transform have shown the benefits of refining the methodology with semi-geostrophic coordinate adjustment (Wolf and Wirth, 2015, 2017) and even using wave activity flux (Takaya and Nakamura, 2001) for RWP diagnostics and optimizing the envelope's shape. Our approach to describe the phase and envelope propagation of a RWP in Fig. 6, with the only use of relative pressure anomalies from gridded GNSS-RO and the Hilbert transform over a limited wavenumber range (5-8), is quite simplistic in comparison but enough to get a good qualitative view. We concentrate now on the first RWP in Fig. 6 (2009-02-27 to 2009-03-07), exploring how its pressure and temperature structures evolve in the UTLS. Longitude-height snapshots of this RWP are presented in 2-day intervals in Figure 7.

The RWP pressure anomalies in Fig. 7 (top row) are centered around the tropopause level, while the temperature anomalies (bottom row) have a clear separation between the UT and LS, and a noisier appearance. This is in agreement with the results from the zonal mean perspective from Figs. 1 and 2. The RWP pressure anomalies have a near-global coverage near the tropopause at almost all times, despite having segments of very low amplitudes: e.g. $\sim$60°E in 2009-03-01 and $\sim$120°E in 2009-03-03 in Fig. 7, top row. The zonal extent of the RWP pressure anomalies decreases away from the tropopause: e.g. compare pressure anomalies at 10 and 16 km in Fig. 7. The choice of a specific vertical level to extract the RWP's envelope could affect the extent of the diagnosed RWP (see discussion in section 5).





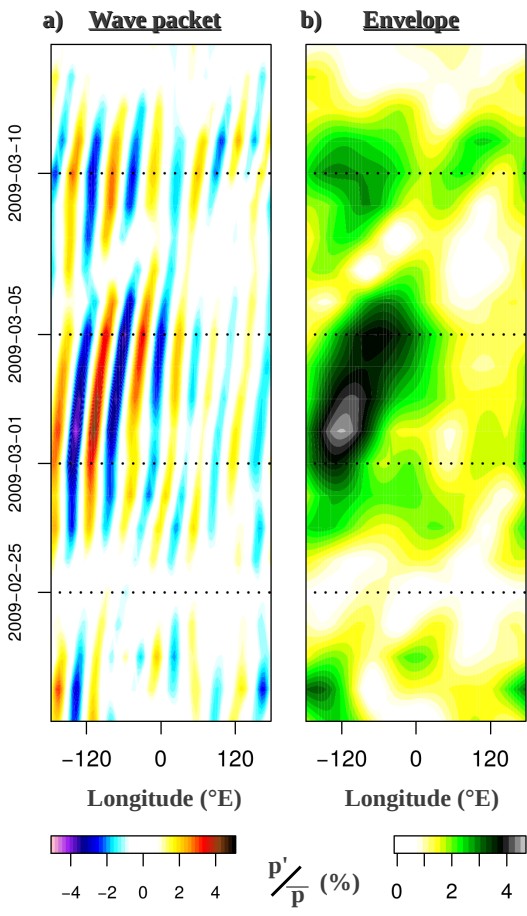

**Figure 6.** (a) Hovmöller diagram of RWP pressure anomalies (w5-8) at zonal mean tropopause level from the end of February to the beginning of March 2009, averaged for 40-60°N. (b) Corresponding envelope of the RWP anomalies on the left side.

Looking at the individual phases of the RWP pressure anomalies, the same troughs and ridges can be recognized going from 6 km up to ∼20 km height in Fig. 7 (top row). A slight westward tilt with height can be spotted in the stratosphere, which is typical of Rossby waves, although the RWP pressure structures in the UTLS can be considered almost barotropic. From the RWP pressure anomalies in Fig. 7 (top row) it can be concluded that RWPs form a direct dynamical connection between the

5   UT and the stratosphere.

The RWP temperature anomalies in Fig. 7 (bottom row), apart from the break around tropopause level, show an out of phase behaviour between the UT and LS: troughs (blue phases in Fig. 7, top row) correspond with negative temperature anomalies in the troposphere and positive temperature anomalies in the stratosphere. This is expected from the meridional advection associated to the pressure anomalies and the inversion of the meridional temperature gradient between the UT and LS (V.

10   Wirth, personal communication). However, the meridional temperature gradients in the lowermost stratosphere are of low





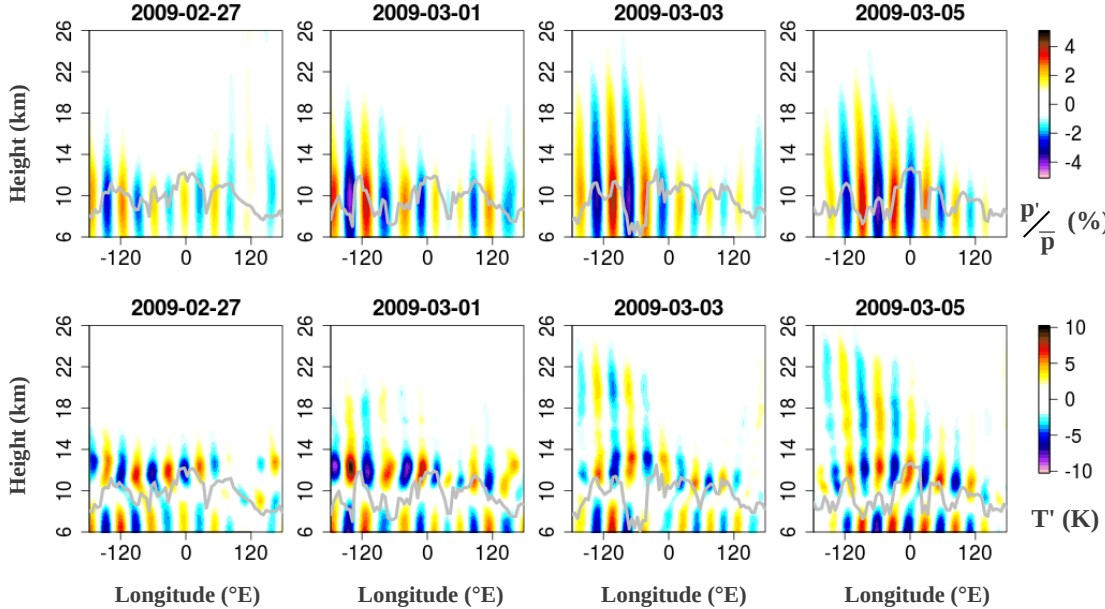

**Figure 7.** Longitude-height snapshots of (top row) pressure and (bottom row) temperature anomalies of the RWP from Fig. 6 at 50°N. The grey line denotes the gridded lapse-rate tropopause height.

magnitude, so it is not clear why the RWP temperature anomalies maximize so close to the tropopause and not higher up. Pilch Kedzierski et al. (2017) also reported wave amplitudes maximizing within a short height range close to the tropopause.

The longitude-height structures of RWP pressure and temperature anomalies in the UTLS from Fig. 7 show a high resemblance with those reported by Hakim (2005) (e.g., see their Fig. 7), although they used the leading empirical orthogonal

functions (EOFs) of meridional winds and temperature. These structures are strongly related to forecast error propagation (Hakim, 2005), with wind errors maximizing near the tropopause while temperature errors maximize near the surface. It is possible to collocate forecast error directly with *observed* RWP structures like those we show in Fig. 7, another interesting topic for a future study.

Recent case studies about extreme Arctic cyclones (Tao et al., 2017a, b) highlighted the influence of strong positive PV

anomalies in the LS with a warm core in the deepening of the surface cyclones, which would be consistent with the presence of an RWP with UTLS pressure and temperature structures such as those depicted in Fig. 7, a high-amplitude trough in this case.

Odell et al. (2013) noted large stratospheric geopotential height tendencies during the 1993 Braer storm, the deepest extratropical low on record with 914 hPa minimum surface pressure. A study of 60 severe European windstorms (Pirret et al., 2017) found that in 20 cases the stratosphere contributed 10% or more to their deepening. The pressure tendency equation used in

Pirret et al. (2017) used the 100 hPa level (∼16 km) as the upper boundary, with the stratospheric contribution being dependent on geopotential height tendencies on this boundary (their d*Phi* term). The presence of a high-amplitude RWP in the UTLS can significantly affect geopotential anomalies at the 100 hPa level, as our results show (Fig. 7, top row; also Fig. 2 indicates





pressure anomalies reaching this level very often), therefore we hypothesize that the d*Phi* term could be very sensitive to the presence of RWP structures in the UTLS.

The out of phase behaviour of RWP temperature anomalies between the UT and LS results from the dynamically barotropic structure of the RWP's troughs and ridges across the UTLS (Fig. 7), the opposing sign of meridional temperature gradients

in the troposphere and the LS, and to some process that amplifies the temperature anomalies close to the tropopause. We will attempt to confirm whether this is a RWP behaviour that happens at all times in section 4.3, with climatological statistics.

### 4.2   The 2010 Moscow heatwave case

The previous section 4.1 showed a typical example of a winter RWP and the evolution of its pressure and temperature structures in the UTLS. Recent studies have shown that in NH summer and with a specific setting of the waveguide(s), resonant conditions

appear and drive high amplitude waves which become quasi-stationary, with wavenumbers of the w6-8 range (Petoukhov et al., 2013; Kornhuber et al., 2017, 2019). This process has been linked to the 2003, 2010, 2015 and 2018 heatwaves in Europe and Russia. Is the appearance of RWPs in the UTLS any different under these special conditions? Throughout this section we will seek an answer by repeating the diagnostics of section 4.1 for the summer 2010 Moscow heatwave.

Figure 8a shows the evolution of RWP pressure anomalies (w4-8) during summer 2010, and Fig. 8b their corresponding

envelope calculated with the Hilbert transform. It can be observed in Fig. 8b that the RWP envelope goes one and a half times around the globe between June 10th and July 10th. Eastward phase propagation is generally visible throughout this period, except for the time when the RWP is around 0° longitude, where the phases become nearly stationary (Fig. 8a). Between July 10th and August 10th, a trough (blue) sets in around 0° and barely moves, with ridges constantly present to its West and East. The ridge around ∼30-45°E is present nearby Moscow during this time. It is not until August 20th when a trough passes over

Moscow's longitude.

Rather than one RWP expanding and becoming stationary, the envelope evolution in Fig. 8b suggests recurrent Atlantic RWPs with eastward group propagation terminating over 0-60°E, with near-zero phase speeds over this region between July 10th and August 10th. The amplitudes of the multiple RWPs in Fig. 8 are not higher than those of the typical winter case in Fig. 6, meaning that RWP anomalies in the UTLS need not be very strong to cause extreme events on the surface. The summer

2010 case is unusual for the low phase speeds around 0° longitude and the recurrence of RWPs coming into the 0-60°E region, giving the event a much longer duration.

In Figure 9 we present longitude-height snapshots of the summer 2010 RWPs. Given the longer event duration, we take a total of 8 snapshots at 4-day intervals. RWP pressure anomalies in the UTLS (1st and 3rd rows) are maximized near the tropopause level and quasi-barotropic, while the RWP temperature anomalies (2nd and 4th rows) maximize above the tropopause with a

clear break between the UT and LS signals. No qualitative differences are found in the UTLS structures of RWPs between the typical winter case (Fig. 7) and the recurrent RWPs with stationary phases during the summer 2010 Moscow heatwave (Fig. 9). A similar analysis of longitude-height snapshots during the 2015 European heat wave leads to the same qualitative conclusions (not shown) in terms of RWP pressure and temperature structures in the UTLS.





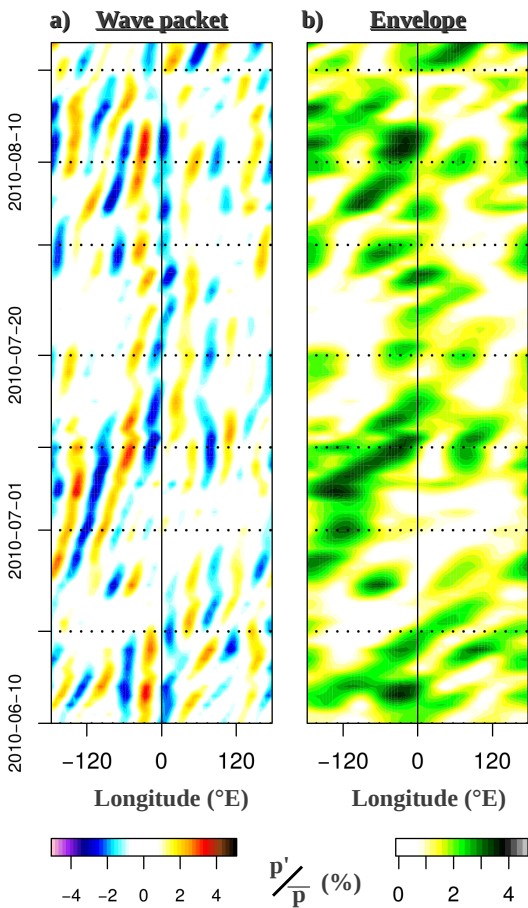

**Figure 8.** (a) Hovmöller diagram of RWP pressure anomalies (w4-8) at zonal mean tropopause level for June-July-August 2010, averaged for 50-60°N. (b) Corresponding envelope of the RWP anomalies on the left side. Horizontal dotted lines mark the $1^{st}$, $10^{th}$ and $20^{th}$ days of the month.

We conclude that the general behaviour of RWPs across the UTLS, dynamically and thermally, is the same in normal winter conditions (Figs. 6 and 7 in section 4.1) and summer under quasi-resonant waveguide setting (Figs. 8 and 9 in section 4.2). The only differing factors during the summer 2010 heatwave are the low RWP phase speeds and their recurrence, although their UTLS appearance is alike to typical winter RWPs.



**Figure 9.** Longitude-height snapshots of (1st and 3rd rows) pressure and (2nd and 4th rows) temperature anomalies of the RWP from n Fig. 8 at 55°N. The grey line denotes the gridded lapse-rate tropopause height.





### 4.3 Climatological statistics of RWP longitude-height structures

**Vertical scale analysis**

Figs. 7 and 9 showed the difference between the RWP appearance in terms of pressure and temperature anomalies: whereas pressure anomalies have a long vertical wavelength, temperature anomalies show a break around the tropopause and opposite

phases in the UT and LS. Therefore it is expected that the vertical scale of RWP temperature anomalies is significantly shorter than that of RWP pressure anomalies, especially in the UTLS. To quantify this, we take all longitude-height snapshots of RWP (w4-8) anomalies between 40-60° latitude, computing Fourier power spectra in the vertical direction between 6-36 km height. The resulting average Fourier power spectra for both hemispheres are shown in Figure 10a.

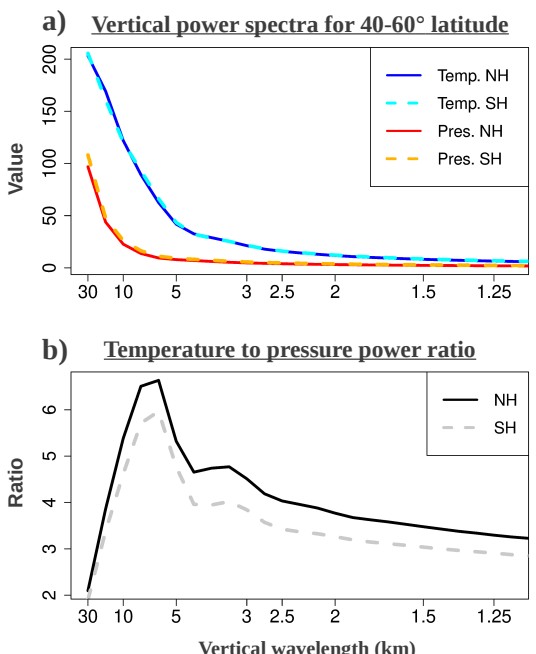

**Figure 10.** (a) Vertical power spectra of all RWP (w4-8) longitude-height snapshots between 40-60° latitude for both hemispheres. (b) Ratio of temperature and pressure power spectra.

Temperature (solid blue, dashed cyan lines) and pressure (red, dashed orange lines) vertical power spectra for NH and

SH show very little interhemispheric differences in Fig. 10a. Both RWP temperature and pressure anomalies (w4-8) have increasing power towards longer vertical wavelenghts. The pressure power spectra show the steepest increase in power for vertical wavelengths >10 km, with a still noticeable amount of power between 5-10 km vertical wavelengths. Temperature vertical power spectra show the steepest increase in power at wavelengths >5 km and a noticeable amount of power between 3-5 km wavelengths. Also, the integrated power of 1.5-3 km vertical wavelengths might be of significance in the temperature

spectrum. Unlike pressure, the temperature structures of RWPs have a relatively large amount of power at vertical wavelengths





3-10 km in their Fourier spectra, and those are the vertical wavelengths where the power ratio between temperature and pressure is highest in Fig. 10b.

In an atmosphere model, the vertical resolution (or the separation between model levels) needed to resolve wave temperature structures is ∼6 times less than the wavelength. Most global climate models (GCMs) have vertical resolutions coarser than 1

km in the UTLS, meaning that they will struggle to represent vertical wavelengths shorter than 6-8 km. From the temperature vertical power spectra in Fig. 10a one could anticipate that RWP temperature structures in the UTLS such as Figs. 7 and 9 may be partially under-represented by GCMs. In fact, GCMs, reanalyses and forecast models are known to produce too smooth temperature gradients in the tropopause region (Gettelman et al., 2010; Hegglin et al., 2010; Birner et al., 2006; Pilch Kedzierski et al., 2016b). The tendency of forecast models of smoothing the tropopause with lead time (Gray et al., 2014)

is partially compensated by diabatic and parameterized processes in the modelled RWPs (Saffin et al., 2017).

A better understanding of RWP diabatic processes offers the chance of forecast improvement, therefore a variety of frameworks to analyze RWP evolution have been developed, mainly using reanalysis or forecast model output (e.g., Hoskins et al., 1985; Orlanski and Katzfey, 1991; Nielsen-Gammon and Lefevre, 1996; Chang, 2001; Gray et al., 2014; Teubler and Riemer, 2016). The vertical structure of such processes is of high importance for RWP evolution, and analyses such as our Figs. 1, 5, 7,

9 or 10 would enable a comparison of model/reanalysis output directly with GNSS-RO observations. See section 5 for a more detailed discussion.

**UT and LS out of phase temperature behaviour**

Figures 7 and 9 indicate that the RWP thermal structures in the UT and LS have constantly opposing phases. If we take the zonal structures of RWP temperature anomalies 1.5 km above and 3 km below the zonal mean tropopause as proxies for the

LS and UT, respectively, in the case of Figs. 7 and 9 the correlation factors between the LS and UT would be close to R = -1. Is this a typical RWPs behaviour? We test this by creating probability density functions (PDFs) of the UT-LS correlation factor R against the wave's amplitude (as the standard deviation of the wave pressure anomalies at zonal-mean tropopause level) for different wavenumber ranges and latitude bands in Figure 11.

The PDF for mid-latitude RWPs (w4-8, Fig. 11a) shows a general tendency for UT-LS anticorrelation of the temperature

structures: higher densities (yellow-red colours) are increasingly packed towards R = -1 the higher the RWP pressure amplitude gets. This confirms that the observed RWP temperature structures shown in the two case studies (Figs. 7 and 9), with opposite phases between the UT and LS, are a typical RWP behaviour with very few exceptions as one can guess from the near-zero probability densities away from R = -1 in Fig. 11a.

Mid-latitude planetary waves (w1-2, Fig. 11b) show a much more disperse distribution resulting in lower densities (grey-

black colours) covering the whole R range, evenly distributed especially at higher wave amplitudes. In Fig. 11b, the relative concentration towards R = -1 at lower wave amplitudes could be the result of the w1-2 components being part of a RWP, as explained next. One may consider a RWP with carrier w4 (e.g., the idealized example in Wolf and Wirth (2015), their Fig. 2d) and its zonal wavenumber Fourier spectrum: the RWP wavenumber spectrum peaks at w4, with decreasing power of the

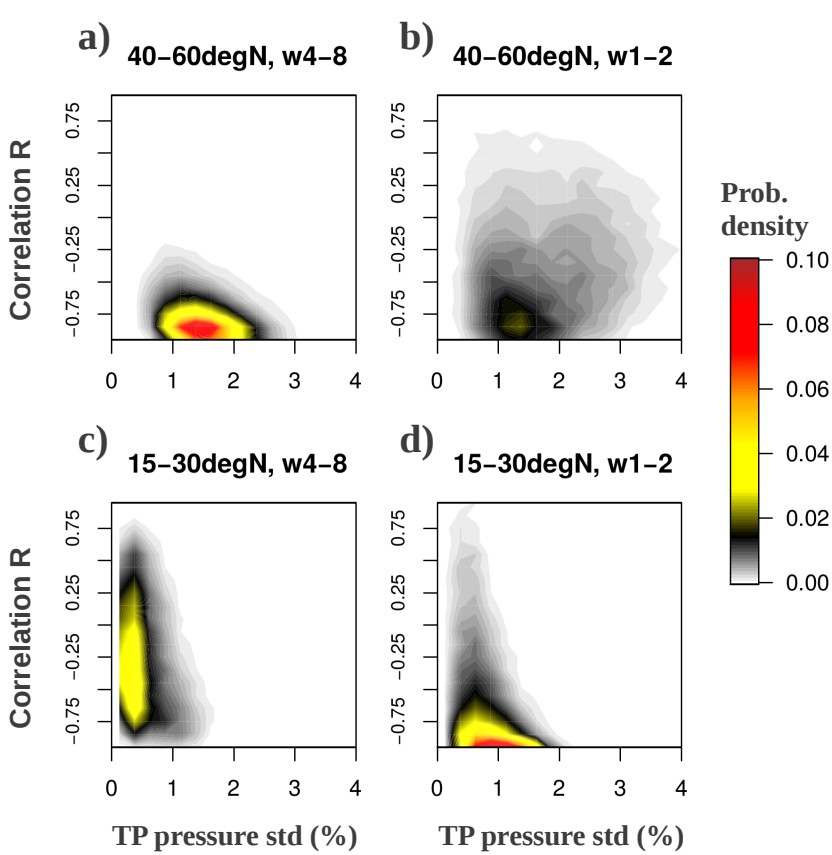

**Figure 11.** Probability density functions of the UT-LS temperature correlation -vs- wave amplitudes for RWPs (w4-8, left column) and planetary waves (w1-2, right column) at mid-latitudes (top row) and the subtropics (bottom row).

wavenumbers around it, and w1-2 are low amplitude contributors to the RWP zonal structure the same way as w6-7. Higher amplitude planetary waves, which cannot be part of RWPs, do not show such a tendency of the PDF to maximize near R = -1.

In the subtropics (Fig. 11c-d), both RWPs and planetary waves have lower amplitudes than their mid-latitude counterparts. This is expected since Figs. 3 and 4 show very little transient (2-20 day period) wave activity there, and any wave amplitude of importance in Fig. 11c-d, which has not been filtered in the time dimension, should therefore come from stationary waves. The PDF for subtropical RWPs (w4-8, Fig. 11c) is concentrated at low amplitudes and shows very little shift towards R = -1, being rather spread throughout all R values and looking nothing like the mid-latitude PDF.

Planetary waves in the subtropics show a PDF packed near R = -1 (w1-2, Fig. 11d). As mentioned above, higher amplitudes of w1-2 belong to stationary signals, which translates into large-scale and long-lasting temperature anomalies of opposite sign in the subtropical UT and LS. A plausible explanation for Fig. 11d are ENSO-related temperature anomalies in the UT and LS: they extend to the subtropics, are long-lasting, and their zonal scale fits well with a combination of zonal wavenumbers 1-2.





Domeisen et al. (2019), their Fig. 5, indeed showed opposite-signed UT and LS temperature regressions to the ENSO index. UT warm anomalies relate to increased convective latent-heat release and LS cold anomalies from the resulting enhanced large-scale upwelling. These anomalies would combine into the anticorrelation shown in Fig. 11d.

We conclude that the UT-LS anticorrelation of temperature anomalies is a general characteristic of RWP longitude-height structures in mid-latitudes (Fig. 11a). However, stationary planetary waves (w1-2) in the subtropics show a similar UT-LS temperature anticorrelation which we link to ENSO-related convection (Fig. 11d). We repeated the analysis of Fig. 11 on the SH, finding nearly identical results (see Fig. S4 in the supplement).

**Zonality of RWP envelopes: hemispheric comparison**

    Early studies noticed the tendency of RWP envelopes to have a near-global scale in the SH, while the NH shows a more

zonally confined behaviour (see Lau (1979); Randel and Stanford (1985) and references therein). Hemispheric comparisons by Chang (1999) and Souders et al. (2014b) with more advanced methods to define RWPs support these earlier findings. A Fourier analysis of the RWP envelope describes its zonal scale: the w0 component of the envelope corresponds to the global (or zonal mean) extent, while w1 and higher components modulate how the envelope's amplitude varies regionally. Here we want to find out whether the SH has higher amounts of all envelope's Fourier components, or only of the longest ones. A comparison of the

wavenumber power spectra of RWP envelopes from NH and SH mid-latitudes is shown in Figure 12.

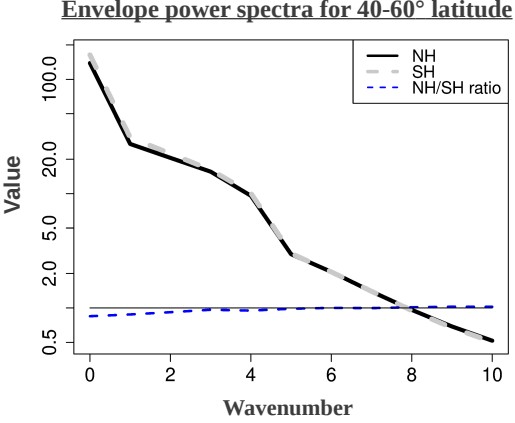

**Figure 12.** Average zonal wavenumber power spectra of RWP envelopes between 40-60° latitude for NH (black) and SH (grey). The blue line denotes the NH/SH ratio, with the thin black line showing a 1/1 ratio. As in Figs. 6 and 8, RWP envelopes are calculated at zonal-mean tropopause level.

    A surprising similarity of the RWP envelope's wavenumber spectra for both hemispheres can be observed in Fig. 12 (solid black and dashed grey lines, also note the logarithmic y-scale). There is a steady exponential increase in power from w10 to w5, appearing as a line of constant slope in Fig. 12 between values from 0.5 to 2. The increase in power from w4 to w1 is well above exponential, with a jump-like shift to values from 10 to ~30. The w0 component has the highest power (~150),

which is expected because a RWP envelope is by definition always positive. It can be concluded that at tropopause level,





w0-4 components markedly dominate the RWP envelope spectrum, with higher wavenumbers contributing marginally. This is observational support to previous studies which filtered out higher wavenumber components of the RWP envelope in their methodology: e.g. Wolf and Wirth (2015) eliminated the envelope's components >w6 to avoid undesired small-scale wiggles in the envelope's shape.

To highlight where the differences between hemispheres are, we add the NH/SH power ratio for each wavenumber as the dashed blue line in Fig. 12. It can be observed that for w3-10, the NH/SH ratio is very close to 1: it ranges from 0.95 to 1.03. The NH/SH power ratio becomes lower at larger scales: 0.92 for w2, 0.88 for w1 and 0.85, the biggest hemispheric difference, for w0. This means that the longest wavenumber components (w0 and w1) of RWP envelopes in the NH climatologically have a 85℅ and 88℅ of the power found in the SH spectrum. Our results in Fig. 12, with the SH having more w0 and w1 in the RWP

envelopes, are consistent with previous studies (Lau, 1979; Randel and Stanford, 1985; Chang, 1999; Souders et al., 2014b) in that the zonal extent of RWPs in the SH tends to be more global than in the NH. In addition, we show that the amount of zonal variability of the envelope's amplitudes at sub-planetary scale (w2 and higher) is basically the same in both hemispheres.

## 5   Discussion: on the importance of following the tropopause

### a) for diagnosing RWP properties

In section 3 we showed time-height sections of zonally averaged RWP activity in Figs. 1 and 2 and the climatological vertical profiles of RWP activity in Fig. 5. Meanwhile in section 4, longitude-height snapshots of RWP anomalies were shown in Figs. 7 and 9. All of these Figures depict RWP activity being centered around tropopause height, following it at all times. The RWP pressure anomalies in our study, defined relative to the zonal-mean pressure at each level (let us name it $P'_{rel} = P'/\bar{P}$), would highly correlate with meridional winds since they are conveniently proportional to geostrophic winds as explained next.

Under geostrophic balance: $fv_g = \frac{1}{\rho} \cdot \frac{\delta P}{\delta x}$ with $f$ the Coriolis parameter, $v_g$ geostrophic meridional wind, $\rho$ air density, $P$ pressure and $\delta/\delta x$ its partial derivative in the longitude dimension. Knowing $\rho = P/RT$ where $R$ is the specific gas constant for air and $T$ absolute temperature:

$$v_g = \frac{RT}{f} \cdot \frac{\delta P}{P\delta x} \quad \propto \quad \frac{\delta P'_{rel}}{\delta x} \tag{4}$$

Note that both $T$ and $P$ on equation 4 are in absolute terms, with zonal variations relatively small and therefore influencing

$v_g$ much less than the $\frac{\delta P}{\delta x}$ term. It follows that wave amplitudes of $v_g$ and $P'_{rel}$ should remain proportional over a given height range, and our results (e.g., Figs. 2 and 5b) indicate that $v_g$ will decrease away from the tropopause, in agreement with previous studies. At mid-latitudes, the tropopause in the NH varies from ∼220-300 hPa in winter to ∼175-250 hPa in summer; while in the SH it varies from a rather meridionally uniform ∼250 hPa in winter to ∼200-280 hPa in summer (Son et al., 2011). This has important implications for RWP diagnostics, which are usually performed with meridional winds at a fixed pressure level,

which is not equidistant from the tropopause over latitude and season.

    The most widely used pressure level for RWP diagnostics is 300 hPa, either for showcasing RWP envelope reconstruction/tracking methods (Zimin et al., 2003, 2006; Souders et al., 2014a; Wolf and Wirth, 2015, 2017), producing RWP clima-



tologies/composites (Chang, 1999; Chang and Yu, 1999; Williams and Colucci, 2010; Souders et al., 2014b) or studying their predictability (Quinting and Vitart, 2019). The 250 hPa level was also used by Glatt and Wirth (2014) and Grazzini and Vitart (2015) to study RWP properties and predictability in forecast models.

    Generally a fixed threshold for RWP envelope amplitude is defined to determine whether a RWP is present and delimit the
RWP's horizontal extent. Having the longitude-height snapshots of RWP anomalies from Figs. 7 and 9 in mind, the diagnosed RWP amplitude and extent would increase the closer the chosen vertical level is to the zonal-mean tropopause. For a given pressure level, its distance from the tropopause level will vary seasonally and over latitude, introducing an aliasing factor if RWP properties are compared between summer/winter or NH/SH. Sometimes this issue could even affect the number of diagnosed RWPs: if the pressure level is further from the tropopause than usual, some part of a RWP may show meridional
winds below the threshold which leads to fragmentation into two or more RWPs; or a low-amplitude RWP may not be detected at all while at tropopause level the threshold is still exceeded. Meridional wind on the 2 PVU surface (the so-called dynamical tropopause) is an available reanalysis product that is a perfect candidate to avoid the above-discussed aliasing effect on RWP properties.

    Some methods make use of a threshold relative to the zonally averaged RWP envelope amplitude (e.g., Glatt and Wirth, 2014;
Grazzini and Vitart, 2015; Quinting and Vitart, 2019). While solving the issue of pressure level distance to the tropopause, in this case an even bigger problem arises: the presence of stronger wave activity over a near-global longitude range would increase the threshold substantially, reducing the diagnosed RWP size (Wolf and Wirth, 2017). Also, in the case of a zonally localized RWP growing into a high-amplitude and global wave due to resonant conditions (e.g., Petoukhov et al., 2013), a relative threshold would on the contrary yield a shrinking RWP.

**b) for RWP energy budgets and fluxes**

    RWP eddy kinetic energy (EKE) budgets are usually presented as vertical integrals, from the surface up to the upper boundary at 100 hpa (Orlanski and Katzfey, 1991; Chang, 2000, 2001) or 50 hPa (Orlanski and Sheldon, 1993). Additionally, Chang (2001) showed volume-averaged composites of RWP wave activity fluxes for the UTLS (400-200 hPa), noting that the baroclinic growth – barotropic decay paradigm of RWP life-cycles has to be treated with caution when interpreting zonally localized
waves. PV inversion diagnostics by e.g. Nielsen-Gammon and Lefevre (1996) use 1000 and 100 hPa as boundaries for the PV tendency equation, presenting different terms and compositing at a fixed pressure level (e.g. near 300 hPa).

    Our results suggest that compositing energy budget terms and fluxes relative to the tropopause height, or even making separate budgets for the UT and LS, would add precision to the magnitude, location and role of each term. Wind-dependent terms would maximize near the tropopause, since our RWP pressure anomalies are proportional to geostrophic winds. The
temperature-dependent terms would show high sensitivity to their location relative to the tropopause, since our RWP temperature signals maximize right above the tropopause and become very low right below it (see Figs. 1, 2, 5, 7 and 9). Averaging on pressure levels may lead to smoothing of the vertical structures of specific EKE budget or PV tendency terms. Tropopause-based averaging retains UTLS gradients as introduced by Birner et al. (2002) and Birner (2006) for radiosonde temperature structures.





Recent publications are already applying a tropopause-relative framework to study PV tendencies (e.g., Cavallo and Hakim, 2009; Saffin et al., 2017), compositing for cyclones and troughs/ridges. Such an approach combined with a refined method to localize RWP envelopes (see discussion above) could improve the understanding of the forcings involved in RWP life-cycles and their interaction with the background flow, e.g. in 3D composites similar to those presented by Chang (2001).

**c) for diagnosing wave trends**

Quantifying the waviness of the extratropical westerly flow is challenging (see review by Coumou et al. (2018)): each measure may have a different physical meaning and different degrees of complexity for systematic implementation on reanalysis or model data. Waviness trends are inconsistent among different methodologies (e.g., Barnes, 2013; Coumou et al., 2015; Cattiaux et al., 2016). These studies have in common that they perform their analyses on fixed pressure levels (e.g., 500 and 250 hPa for wave amplitudes (Barnes, 2013; Cattiaux et al., 2016), or 850-250 hPa integrals for EKE trends (Coumou et al., 2015)). Thus, they do not consider tropopause height biases in the models used for climate projections, which can be different for each model, potentially adding spread to the results. Neither do these studies account for modeled or reanalysed tropopause height trends (e.g., tropospheric expansion leads to increase of extratropical tropopause height over time with climate change). Interannual variability of the extratropical tropopause height alone would introduce aliasing effects over the diagnosed waviness on a fixed pressure level. Wave amplitudes diagnosed on the 2 PVU surface, as our results show, would always be equidistant from the location of the maximum in wave activity in the jet stream, thus providing the fairest trend measure and model intercomparison in terms of wave amplitudes.

The issue of tropopause trends was already noted for the attribution of stratospheric residual circulation trends by Oberländer-Hayn et al. (2016). We add that, since RWPs show a quasi-barotropic structure throughout the UTLS (See Figs. 7 and 9), RWP phase *speeds* in principle are not subject to the aliasing effect of varying tropopause distance from the diagnosed pressure level, so phase speed trends like those studied by Coumou et al. (2015) are not affected by this issue.

## 6 Conclusions

Our study is a first attempt to describe RWP properties globally with the sole use of GNSS-RO observations, focusing on both hemisphere's extratropical UTLS. Observational knowledge about RWPs such as the one presented here is much needed for comparison and interpretation of climate model and reanalysis output, especially with the current scientific effort to better understand storm track and jet stream responses to climate change (e.g., Hall et al., 2015; Shaw et al., 2016). Our results are relevant for stratosphere-troposphere coupling: they indicate systematic RWP activity propagation into the stratosphere, which markedly increases during SSWs, in addition to the well-known RWP driving of tropospheric weather. We summarize our main findings below:

1. RWP activity follows the zonal-mean tropopause level and maximizes around it during all seasons (Figs. 1, 2, 5, 7 and 9). RWP pressure anomalies tend to be centered at the tropopause, with decreasing amplitude away from it. RWP temperature anomalies maximize right above the tropopause, with a contrasting minimum right below the tropopause.





2. We note frequent RWP propagation into the stratosphere, sometimes beyond 20 km height in the NH, which mostly manifests for wavenumbers 4 and 5 (Figs. 1 and 2). Enhanced vertical propagation of RWP activity coincided with the SSWs of 2009, 2010 and 2013 (see Figs. S1, S2 and S3). We will explore this further in an upcoming study, given the importance of the lower stratosphere in modulating the onset and downward propagation of SSWs (Karpechko et al., 2017; Domeisen et al., 2018).

3. Since RWP activity constantly follows the tropopause, we note that the use of fixed pressure levels for RWP diagnostics or wave trends (e.g. typically 300 hPa) may induce some aliasing in the resulting quantities, since it is not equidistant from the tropopause over time and latitude. We suggest using the 2 PVU surface to avoid this (see discussions in section 5).

4. In addition to gridded temperature, our study is the first to analyse GNSS-RO (relative) pressure anomalies, whose signals show much better continuity and coherence in the UTLS and the stratosphere (Figs. 1, 2, 7 and 9). This new approach shows many benefits for studying extratropical wave propagation with GNSS-RO data, especially in the vertical direction.

5. The dynamical and thermal appearance of RWPs in the UTLS is generally similar in different seasons and waveguide settings: no qualitative differences were observed after comparing a typical winter RWP with RWP activity during the 2010 Moscow heatwave where resonant conditions were present (Kornhuber et al. (e.g., 2017) see sections 4.1 and 4.2).

6. Overall, RWPs in the SH show a preference for lower carrier wavenumbers compared to the NH (Figs. 3 and 4). In terms of envelope properties, the SH shows a higher amount of the longest scales compared to the NH (w0, w1, Fig. 12). This is in good agreement with previous observational (e.g., Lau, 1979; Blackmon and White, 1982; Randel and Stanford, 1985; Chang, 1999) and model (e.g., Gall, 1976; Simmons and Hoskins, 1978; Hayashi and Golder, 1983) studies. Apart from the above, we found that RWP properties in the UTLS are generally very similar across hemispheres.

*Code and data availability.* The data sets used for this publication were downloaded from the following webpages upon registration: ERA-Interim data http://apps.ecmwf.int/datasets/data/interim-full-daily/levtype=pl/ and GNSS-RO data from different satellite missions http://cdaac-www.cosmic.ucar.edu/cdaac/products.html. The 'R' mathematical software is available here: https://cran.r-project.org/. The 'NCL' space-time filter to extract waves is available here: https://www.ncl.ucar.edu/Document/Functions/User_contributed/kf_filter.shtml.

*Author contributions.* RPK designed the methodology, performed the analyses, produced all Figures and wrote the manuscript. KM and KB contributed with ideas and commented the manuscript.

*Competing interests.* The authors declare that they have no conflict of interest.





*Acknowledgements.* This study was funded by the GEOMAR Helmholtz Centre for Ocean Research in Kiel. We thank the ECMWF data server for the freely available ERA-Interim data. We also thank UCAR for the reprocessing and availability of GNSS-RO data from all satellite missions. Discussions and commenting in the early stages of the study with Dr. Tim Kruschke, Prof. Volkmar Wirth and Prof. Daniela Domeisen are deeply appreciated.



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
