# Peer review of "New insights into Rossby wave packet properties in the extratropical UTLS using GNSS radio occultations"

_Atmospheric Chemistry and Physics, 2020_

## Referee Comment (RC1) · Anonymous Referee #1 · 30 Mar 2020

The present study gives brought insights into the climatological structure and evolution of Rossby wave packets in the middle atmosphere, and potential anomalous behaviour during and before specific Eurasian heat wave and stratospheric sudden warming events. Overall the paper is well written and the scientific content is interesting. In the following are various comments on general and specific aspects of the study:

The paper is generally quite long, which is OK considering that is covers a wide range of topics. However, although I do not have specific examples in mind, I suppose there would also be some scope for shortening.

You mention several times that the RWPs 'defy the Charney-Drazin criterion' as they

reach heights above 20km. This would require the corresponding waves to be vertically propagating, but in most of your plots (e.g. Fig. 1 or Fig 2, keeping in mind the normalisation) it appears to me that the signal is decaying quickly with height and signs of vertical propagation are hard to see.

In your analysis of the 2010 Moscow heat wave you state that 'RWP in the UTLS need not be very strong to cause extreme events on the surface'. How established is the causality that UTLS processes have determined the Eurasian surface fields during that year?

At various places you discuss the 'co-amplification' of the UT and LS signals of RWPs. Considering the almost perfect (anti-)correlation you found, is it sensible to think of it as two separate structures that interact compared to UT and LS signatures of one UTLS structure?

I felt some of the conclusions in section 6 could be slightly more specific. Point 3, based on section 5, claims a potential importance of tropopause-relative frameworks when analysing RWPs. It would be nice to have and idea how significant the use of a non-relative framework can influence certain results. Correspondingly it would be nice to have one or two examples listed for point 4 emphasising the advantages for analysing GNSS-RO data.

---

## Short Comment (SC1) · 15 Apr 2020

Dear authors,

This SC is not intended to provide a detailed review of the manuscript, I would rather comment on some methodological aspects and ask you for clarification.

First, I would like to point out that the utilization of wet profiles for your analysis does not completely support your claim of superiority of your results over the usage of reanalysis data. Note that the wet retrieval typically relies on a 1D-Var method, which needs assimilation of a background information (ECMWF forecast). In the UTLS, which is

the focus region of your manuscript, dry profiles (pure observations) are considered to be of sufficient accuracy (e.g. Danzer et al., 2014). In a dry retrieval, however, both the temperature and pressure are derived from the density using hydrostatic balance and are therefore dependent quantities (see e.g. Pisoft et al., 2018). In this light, I suspect that the difference between your results for the temperature and pressure data are to a large extent dominated by a different visualization of the anomalies (absolute in [K] versus relative in [%] for pressure). It would be fair to show, how much additional information contain the pressure data over the temperature data (non-hydrostatic processes, water vapor, etc.).

As a second point I noticed that internal gravity waves (GWs) are not mentioned throughout the manuscript. It is well known (see e.g. Fritts and Alexander, 2003) that the GW sourcing, propagation and breaking processes are influenced by Rossby waves (which can be considered a slowly varying background for them) and so it seems plausible that GWs follow the Rossby wave activity and can contribute to what you interpret as Rossby wave packet properties. I see one potentially elegant way how to prove that your results do not contain the GW signal – you can show that the vertical power spectra of your RWP anomalies (e.g. your Fig. 10, but better with log axes) are significantly different from the slope of a saturated GW spectrum. If you can think of a simpler argument to discern the possible GW imprints in your results, I encourage you to provide it in the manuscript.

References: Danzer, J., Foelsche, U., Scherllin-Pirscher, B., and Schwärz, M.: Influence of changes in humidity on dry temperature in GPS RO climatologies, Atmos. Meas. Tech., 7, 2883–2896, https://doi.org/10.5194/amt-7-2883-2014, 2014. Fritts, D. C., and Alexander, M. J. ( 2003), Gravity wave dynamics and effects in the middle atmosphere, Rev. Geophys., 41, 1003, doi:10.1029/2001RG000106, 1. Pisoft, P., Sacha, P., Miksovsky, J., Huszar, P., Scherllin-Pirscher, B., and Foelsche, U.: Revisiting internal gravity waves analysis using GPS RO density profiles: comparison with temperature profiles and application for wave field stability study, Atmos. Meas. Tech., 11, 515–527,

https://doi.org/10.5194/amt-11-515-2018, 2018.

---

## Referee Comment (RC2) · Anonymous Referee #2 · 29 Apr 2020

This study investigates Rossby waves packets, their characteristics and evolution in the extratropical upper troposphere and lower stratosphere, based on observations from radio occultation. Rossby waves packets' structures are analyzed under normal conditions and compared to specific events such as periods of sudden stratospheric warming and heat waves. In connection to the SSW 2009, enhanced vertical propagation of Rossby wave activity is shown. During the Russian heatwave no distinct difference to normal conditions was found. Overall the paper is very informative. It adds new knowledge to the field and discusses ways forward.

Major comments:

[Figure]

The analysis is based on temperature and pressure profiles from radio occultation and I have two major comments in this respect.

1) My first comment relates to the use of temperature from a moist air retrieval and the added value of pressure over temperature. I refer to and fully agree with the comment posted by Petr Sacha, which I do not repeat here.

(i) Dry temperature is recommended favorable for use in the UTLS as it is directly retrieved from refractivity, and does not contain background information from a moist-air retrieval.

(ii) Regarding the added value of pressure, I also strongly recommend to provide further evidence here.

2) Secondly, in your work, you used UCAR CDAAC data. Note, that from bending angle initialization the NCEP climatology might introduce some artefacts in mid to high latitude winter. There is some recent work of the radio occultation community comparing RO data from several processing centers, which discusses this, see Fig.5b to 7b, subpanels 60N –90N, at https://doi.org/10.5194/amt-2019-358).

In your work, however, you perform quality control of the radio occultation profiles (described in section 2.2 of your manuscript), which might get hold of this issue. You state that your QC removes about 10% of the profiles. It would be interesting to see, if there is a pattern in the removed profiles (latitude, time wise), and whether your QC preferably removes profiles in northern high-latitude winter.

Overall, the paper is well written with good discussions and explanations but has a lengthy style and gives repeating information at some places. There is potential for streamlining and shortening at several places to make it better readable without loss of information. I cannot point to each specific place but give examples in the list of minor comments below. I recommend that the authors thoroughly read through the manuscript and try to further streamline it (e.g., remove repeating statements). Regarding the conclusions section: I recommend merging the summary given in section 3 with the conclusion section. I also recommend removing all citations in the conclusions section, all of them have been cited already during the discussions. Please find a list of minor comments below.

Minor comments:

The percentage signs looks strange, is it of different font than the other text?

Use "$\pm$" instead "+-"

P3,l24: Suggest adding a more recent reference on the recent availability of RO data, e.g., Anthes et al. 2011 (https://doi.org/10.5194/amt-4-1077-2011), or Ho et al. 2019 (https://doi.org/10.1175/BAMS-D-18-0290.1)

P8, l1: Please state the time of the SSW, of onset and duration.

P13,l5: "Fig3f: Why is the correlation so high for wavenumbers larger than 8. Only low wave activity at wavenumber 8 is shown in Fig. 1 and 2).

P15, l7: Remove first sentence (it is repeating).

P16 Move the short summary and merge it with the conclusions section.

P16, l27: Remove: "After analyzing RWP activity from a zonally averaged perspective in Section 3, . . ."

P18, end of figure caption: Remove "on left side."

P20, l1: Use "geopotential height tendencies" instead of "dphi"

P20, l3-6: Mover this paragraph up by two paragraphs and insert it in p19, l3, where you discuss the out of phase behavior.

P22, figure caption, end of l1: Remove "n" before "Fig."

P29 &30: remove citations in conclusion section, will make it better readable

P30: l15: "Mosow heat wave": change to "Russian heat wave"

All figures: remove the underline in figure titles.

Figures 10 to 12: make the fonts consistent for x and y titles.

Figures 3, 4: remove "Amp(p')/p" in legend

Figures 6,7: remove "p'/p" in legend and in all other figures. You state it in caption and text.

Fig. 11: use smaller font for a), b), c), d)

---

## Author Comment (AC1) · 15 Jul 2020

The comment was uploaded in the form of a supplement:
https://www.atmos-chem-phys-discuss.net/acp-2020-124/acp-2020-124-AC1-supplement.pdf

---

## Author Response (AR1)

**Response to reviews of Pilch Kedzierski et al.: "New insights into Rossby wave** packet properties in the extratropical UTLS using GNSS radio occultations"**

Dear Editor,

We would like to thank the two anonymous reviewers and Petr Šácha for their helpful comments. In the following paragraphs we include general remarks about common major comments, point-by-point responses to each comment in the reviews along with the manuscript with tracked changes. The referee's comments are in blue font, and our replies are in normal font. Changes made in the revised manuscript are highlighted.

Yours sincerely,

Robin Pilch Kedzierski Katja Matthes Karl Bumke

**General remarks**

We noted two main issues that several reviewer's comments had in common, which we address next.

**(1): On the use of the 'wetPrf' product over dry temperature profiles**

The use of the 'wetPrf' product is not only justified but necessary, keeping in mind the height range used in our study (6-26+ km). The retrieved moisture profile from 'wetPrf' is not considered superior to the analysis used as background (ECMWF). But the retrieved GNSS-RO wet temperature has zero difference with dry temperature above ~10km, while showing advantages at lower levels, as explained in the following paragraph that we added to section 2.1:

The 'wetPrf' product is the best suited for our study for several reasons. In regions where the atmosphere is dry (roughly above 10 km height), the 'wetPrf' and the dry temperature 'atmPrf' profiles basically coincide (Alexander et al., 2014; Danzer et al., 2014). At lower levels, water vapor is increasingly influential and dry temperatures get colder than the real temperature (e.g. at 8 km height in the extratropics, the difference is already of the order of 0.5-2 K and increasing downwards, in Danzer et al. (2014)). Our wave analyses start at 6 km height, and we note that the extratropical tropopause can vary between 7-12 km height over one latitude band on the same day (see sections 4.1 and 4.2). Water vapor content within the RWP's troughs and ridges can be very different at a given vertical level and its effect on the GNSS-RO profiles, especially within the lower part of the UTLS, cannot be neglected if reliable wave amplitudes are desired. To account for water vapor effects on the retrieved GNSS-RO profile, a background state (typically analyses) is assimilated using one-dimensional variational (1D-Var) technique (Healy and Eyre, 2000; Poli et al., 2002). Although this procedure combines the GNSS-RO measurement with the 'first-guess' background, it has been shown that between 9-22 km height the GNSS-RO measurement dominates the retrieved vertical profile (Healy and Eyre, 2000), whereas at lower levels there are temperature improvements from the background towards radiosonde measurements where available (Poli et al., 2002).

**(2): Shortening of repeating information in the manuscript**

- We moved the explanation of how our relative pressure wave amplitudes are proportional to (geostrophic) meridional wind wave amplitudes from section 5 into section 2.4, as we feel this is relevant information that the reader should have in mind when interpreting temperature and pressure RWP anomalies in our plots.

- In section 3, we shortened the first paragraph, as well as removing the short summary at the end of the section.

- We merged and shortened the first two paragraphs of section 4.

- We also shortened the transition between the end of subsection 4.1 and beginning of 4.2.

- In section 4.3, in the part about UT and LS out of phase behaviour of the temperature signals, we removed some sentences in the second and last paragraphs.

Overall, the results+discussion text of the manuscript is about a page shorter. However, note this is offset by the increased amount of detail in section 2 arising from the justification of the use of the 'wetPrf' GNSS-RO product and the aforementioned addition to subsection 2.4. This also added several new items to the reference list.

\_\_\_\_\_

**Point-by-point responses to Anonymous Referee #1**

The paper is generally quite long, which is OK considering that is covers a wide range of topics. However, although I do not have specific examples in mind, I suppose there would also be some scope for shortening.

We shortened and/or rearranged several text parts in the revised manuscript, see details in General remarks (2) above.

You mention several times that the RWPs 'defy the Charney-Drazin criterion' as they reach heights above 20km. This would require the corresponding waves to be vertically propagating, but in most of your plots (e.g. Fig. 1 or Fig 2, keeping in mind the normalisation) it appears to me that the signal is decaying quickly with height and signs of vertical propagation are hard to see.

We added some visual aid to an extract of Figure 2 (for w4) to highlight how the wave activity in the lowermost stratosphere (dark green arrows, delimited by vertical dotted lines) arrives later to the middle stratosphere (light green arrows).

We now make it more clear throughout the manuscript that the relative pressure wave anomalies are proportional to meridional geostrophic wind wave anomalies.

Although decaying with height most times, some burst of equal magnitude near the tropopause reach higher than others (compare beginning of Dec. 2008 and end of March 2009). Also, the wave burst in mid-January 2009 (of lesser magnitude near the tropopause than the other two mentioned before), around the time of the SSW, can be clearly seen propagating up to 26km without any amplitude loss. We highlight these points more in the text while discussing Fig. 2 now.

In your analysis of the 2010 Moscow heat wave you state that 'RWP in the UTLS need not be very strong to cause extreme events on the surface'. How established is the causality that UTLS processes have determined the Eurasian surface fields during that year?

The Kornhuber papers (2017, 2019) mentioned at the beginning of this section diagnosed quasi-resonant amplification of Rossby waves with meridional winds in the upper troposphere (300 hPa).

At various places you discuss the 'co-amplification' of the UT and LS signals of RWPs. Considering the almost perfect (anti-)correlation you found, is it sensible to think of it as two separate structures that interact compared to UT and LS signatures of one UTLS structure?

We agree with the reviewer, the term co-amplification is not the most accurate. As the reviewer points out, it implies some interaction of the two structures involved. In section 4.1 (Fig. 7) we discuss the formation of the temperature anomalies by advection and an inverting meridional T gradient across the UTLS, i.e. without the need of interaction between the UT and LS temperature anomalies.

We removed the following sentence from the second-last paragraph in p.8: "This indicates the existence of co-amplification between the UT and LS temperature signals, in addition to the co-amplification of potential vorticity (PV) anomalies between the surface and UT (Hoskins et al., 1985)."

Throughout the text we no longer use co-amplification, but the term 'simultaneous amplification'. The headers of Figs. 3 and 4 have also been adapted accordingly.

I felt some of the conclusions in section 6 could be slightly more specific. Point 3, based on section 5, claims a potential importance of tropopause-relative frameworks when analysing RWPs. It would be nice to have and idea how significant the use of a non-relative framework can influence certain results. Correspondingly it would be nice to have one or two examples listed for point 4 emphasising the advantages for analysing GNSS-RO data.

To point 3 we added: "*RWPs would lose envelope extent and/or amplitude the further the pressure level is from the tropopause*".

Point 4 now reads: "In addition to gridded temperature, our study is the first to analyse GNSS-RO (relative) pressure anomalies, that are conveniently proportional to geostrophic meridional wind wave amplitudes (see section 2.4). The filtered RWP pressure signals show much better continuity (i.e. no breaks in their vertical structure) and less noisiness in the UTLS and the stratosphere, compared to temperature wave signals (Figs. 1, 2, 7 and 9). This new approach shows many benefits for studying extratropical wave propagation with GNSS-RO data, especially in the vertical direction."

\_\_\_\_\_

**Point-by-point responses to Anonymous Referee #2**

**Major comments:**

The analysis is based on temperature and pressure profiles from radio occultation and I have two major comments in this respect.

1) My first comment relates to the use of temperature from a moist air retrieval and the added value of pressure over temperature. I refer to and fully agree with the comment posted by Petr Sacha, which I do not repeat here.

(i) Dry temperature is recommended favorable for use in the UTLS as it is directly retrieved from refractivity, and does not contain background information from a moist-air retrieval.

(ii) Regarding the added value of pressure, I also strongly recommend to provide further evidence here.

In General remarks (1) we detail why the 'wetPrf' product is the best suited for our study. Regarding the added value of pressure, we specify now throughout the text that we refer to wave anomalies and the better continuity and less noisiness of RWP signals in the vertical direction, which is already evident from Figs. 1, 2, 7 and 9.

2) Secondly, in your work, you used UCAR CDAAC data. Note, that from bending angle initialization the NCEP climatology might introduce some artefacts in mid to high latitude winter. There is some recent work of the radio occultation community comparing RO data from several processing centers, which discusses this, see Fig.5b to 7b, subpanels 60N –90N, at https://doi.org/10.5194/amt-2019-358).

In your work, however, you perform quality control of the radio occultation profiles (described in section 2.2 of your manuscript), which might get hold of this issue. You state that your QC removes about 10% of the profiles. It would be interesting to see, if there is a pattern in the removed profiles (latitude, time wise), and whether your QC preferably removes profiles in northern high-latitude winter.

Thank you for the reference, which we added to the data section (2.1). In p.5, l.9-14 of the revised manuscript we specify that after quality control we find no discontinuities in the data. We added the following statement to this paragraph: "*The RWP temperature anomalies we find after filtering our gridded dataset (see section 2.4) are one-two orders of magnitude higher than the spread of large-scale and long-term temperature differences across different GNSS-RO processing centres, depending on the height range (Steiner et al., 2020).*"

Below we show time-latitude plots with the location of eliminated profiles during quality control steps 1 and 2 (as step 1 takes very long and discarded profiles are not saved, we only show 2 years). There is a clear tendency of the eliminated profiles to be located in extratropical latitudes, but more of them are eliminated in the summer season.

Note that the outliers eliminated by our quality control procedure have temperature profiles in the stratosphere that are either tens of degrees too cold or too warm than the surrounding profiles. Some of the differences shown by Steiner et al. (2020) could arise from a preference for one type of outlier, but this is speculative and we feel a discussion about this in our section 2.1 or 2.2 would not provide any further valuable information for the reader.